# Genome-scale metabolic reconstructions of multiple *Salmonella* strains reveal serovar-specific metabolic traits

Yara Seif[1], Erol Kavvas [1], Jean-Christophe Lachance[2], James T. Yurkovich [1,3], Sean-Paul Nuccio[4], Xin Fang[1], Edward Catoiu[1], Manuela Raffatellu [4], Bernhard O. Palsson[1,3,4,5] & Jonathan M. Monk[1]

*Salmonella* strains are traditionally classified into serovars based on their surface antigens. While increasing availability of whole-genome sequences has allowed for more detailed subtyping of strains, links between genotype, serovar, and host remain elusive. Here we reconstruct genome-scale metabolic models for 410 *Salmonella* strains spanning 64 serovars. Model-predicted growth capabilities in over 530 different environments demonstrate that: (1) the *Salmonella* accessory metabolic network includes alternative carbon metabolism, and cell wall biosynthesis; (2) metabolic capabilities correspond to each strain's serovar and isolation host; (3) growth predictions agree with 83.1% of experimental outcomes for 12 strains (690 out of 858); (4) 27 strains are auxotrophic for at least one compound, including ʟ-tryptophan, niacin, ʟ-histidine, ʟ-cysteine, and p-aminobenzoate; and (5) the catabolic pathways that are important for fitness in the gastrointestinal environment are lost amongst extraintestinal serovars. Our results reveal growth differences that may reflect adaptation to particular colonization sites.

[1] Department of Bioengineering, University of California, San Diego, La Jolla, USA. [2] Département de Biologie, Université de Sherbrooke, Sherbrooke, QC, Canada. [3] Bioinformatics and Systems Biology Program, University of California, San Diego, La Jolla, USA. [4] Department of Pediatrics, University of California, San Diego, La Jolla, CA, USA. [5] Novo Nordisk Foundation Center for Biosustainability, Technical University of Denmark, Kemitorvet, Building 220, 2800 Kongens, Lyngby, Denmark. Correspondence and requests for materials should be addressed to B.O.P. (email: palsson@ucsd.edu) or to J.M.M. (email: jmonk@ucsd.edu)

The genus *Salmonella* encompasses a variety of gram-negative, rod-shaped flagellated bacteria responsible for an estimated 115 million human infections and 370,000 deaths per year[1–5]. To date, most strains of *Salmonella* have been classified into serovars based on the unique combination of their surface antigens[6]. Taken together, serovars of *Salmonella* can infect a wide range of hosts, including humans and other warm-blooded animals, cold-blooded animals, and plants. Most *Salmonella* serovars colonize a broad range of hosts (e.g., *S. enterica* serovars Typhimurium, and Enteritidis); herein referred to as generalists, while relatively few *Salmonella* serovars have adapted to unique hosts (e.g. *S. enterica* serovar Typhi, which only infects humans); herein referred to as specialists[7,8]. Beyond host specificity, serovars of *Salmonella* have also been shown to differ in their antimicrobial resistance profiles[9] and virulence phenotypes[10]. *Salmonella* serovars are commonly classified into typhoidal and non-typhoidal serovars based on whether they can cause systemic illness or localized gastroenteritis, respectively.

This diversity of lifestyles and host-types is reflected in the different *Salmonella* genotypes. Previously, signature genes, metabolic capabilities and nutrient auxotrophies (inability of an isolate to synthesize a nutrient that is essential for its growth) have been exploited to develop assays for classifying and identifying *Salmonella* strains from other bacteria. However, metabolic capabilities between *Salmonella* isolates have not yet been compared systematically based on whole-genome sequences and predicted metabolic capabilities. Strain-specific metabolic network reconstructions have proven to be powerful tools to probe the effect of genomic diversity between strains of *E. coli* and *S. aureus*[11,12]. A curated genome-scale reconstruction of *S*. Typhimurium str. LT2 exists[13] and has been widely used and expanded to successfully predict virulence phenotypes in infected mouse tissue[14].

Here we built strain-specific metabolic reconstructions for the species and subspecies of the *Salmonella* genus using strains with defined serovars and fully sequenced genomes. We selected 410 high-quality, closed-genome sequences of *Salmonella*[15,16], spanning both species, three subspecies and 64 different serovars. Choosing genomes that provide a diverse variety of subspecies, serovars, and hosts allowed for a comprehensive comparison of genomic features and metabolic capabilities across the *Salmonella* genus. Using this set of strain-specific information, we reveal the basis for serovar-specific and host-associated metabolic traits.

## Results

**Characterizing the *Salmonella* core and pan-genomes**. Our first goal was to characterize the core (genes shared among all strains) and pan (the totality of all genes found across all strains) genomes of the *Salmonella* genus. We analyzed 410 gapless genomic sequences of *Salmonella* strains representing both subspecies, 22 serogroups and 64 total serovars (see Methods, Supplementary Fig. 1 and Supplementary Data 1). All genomes were re-annotated to avoid differential gene calling. We found an average number of coding regions of 4441 per genome. For consistency, plasmids were excluded from the study and all annotated genes were assumed to be functional. We subsequently constructed the *Salmonella* pan-genome (Methods), which contains a total of 21,377 gene families, 1705 of which constitute the core genome. We constructed core and pan-genome curves for a randomly sampled subset of strains focused on up to 10 strains from each serovar (Fig. 1a, Supplementary Data 2). In a previously constructed pan-genome of *Salmonella*, 11,443 gene families were identified across 29 genomes, of which 3211 were conserved[17]. The pan-genome curve is shaped by the number of novel gene family additions with each additional genome sequence. Conversely, the core genome curve represents the number of gene families that were consistently observed for each addition of a genomic sequence for a new strain.

The pan and core genome curves demonstrated that more genomic content is shared within strains of a serovar. The large increases in the slope of the pan-genome curve occurred when the genome being added originates from a serovar that was not represented by any of the previously included genomic sequences. In fact, the largest slopes occurred when genomic sequences of different subspecies were introduced. On average, two genomes of the same serovar shared 365 more genes than two genomes of different serovars (unpaired *t*-test, $p < 0.05$). These observations imply that: (1) in addition to the genes encoding antigen biosynthesis[18] (used for serological determination and identification), each serovar has a defined repertoire of protein families that could be reflective of its respective lifestyle; and (2) the number of shared gene families between two *Salmonella* isolates decreases as the phylogenetic distance between them increases. We demonstrated this second observation empirically by computing phylogenetic distances between strains using the concatenation of 7 *Salmonella* housekeeping genes. We calculated the number of gene families that are not shared between pairs of strains (which we used as a second measure of phylogenetic distance) and plotted these distances against the corresponding pairwise phylogenetic distances calculated in the previous step. Applying a linear regression algorithm, we found that the two measures of phylogenetic distances were highly correlated with an $R^2$-value of 0.851 (*p*-value < 0.01). (Supplementary Fig. 2)

**Strains of a serovar illustrate unique pan-genome features**. Having determined that the core and pan-genome curves are descriptive properties of a serovar, we next asked whether these properties are reflective of its host range. To answer this question, we examined the pan and core genome curves for three serovars of *S. enterica* subsp. *enterica* for which we had a sufficient number (>40) of high-quality genomes: *S.* Paratyphi A (specialist; $n = 41$), *S.* Enteritidis (generalist; $n = 159$), and *S.* Typhimurium (generalist; $n = 46$) (Fig. 1b, Supplementary Data 2, Supplementary Table 1). For each curve, we compared the mean and standard deviation of the number of gene families at the 20th genomic addition (20 was selected because it represented a halfway point in the pan-genome curve, Methods). We found that a serovar with a larger range of hosts does not necessarily have a larger pan-genome. *S.* Paratyphi A and *S.* Enteritidis have a similar number of gene families in the pan-genome at addition 20 ($p(20) = 4527 \pm 43$ and $p(20) = 4606 \pm 75$, respectively). Additionally we found that the *S.* Typhimurium pan-genome is as large as the *Salmonella* pan-genome ($p(20) = 6559 \pm 273$ and $p(20) = 7676 \pm 711$, respectively), suggesting that strains of this serovar are a major source of *Salmonella* gene content variation. Thus, the number of gene families in a serovar's pan-genome does not necessarily reflect the number of hosts it can colonize.

The serovar-specific pan-genome content reveals that there is an underlying repertoire of gene families that are conserved among strains of a serovar, some of which are unique to the serovar. A cluster map (Fig. 1c) of gene families contained in the accessory genome (obtained by subtracting the core genome from the pan-genome) of our three selected serovars demonstrates that each serovar is differentiated by a set of conserved gene families (framed in red). Interestingly, while some genes are classified as accessory in the pan-genome of the *Salmonella* genus, they appear in the core genome of a given serovar (Fig. 1d, Supplementary Data 2). Genes involved in metabolic functions were part of these unique serovar-specific core genomes, which motivated us to characterize the differences in metabolic networks of different serovars.

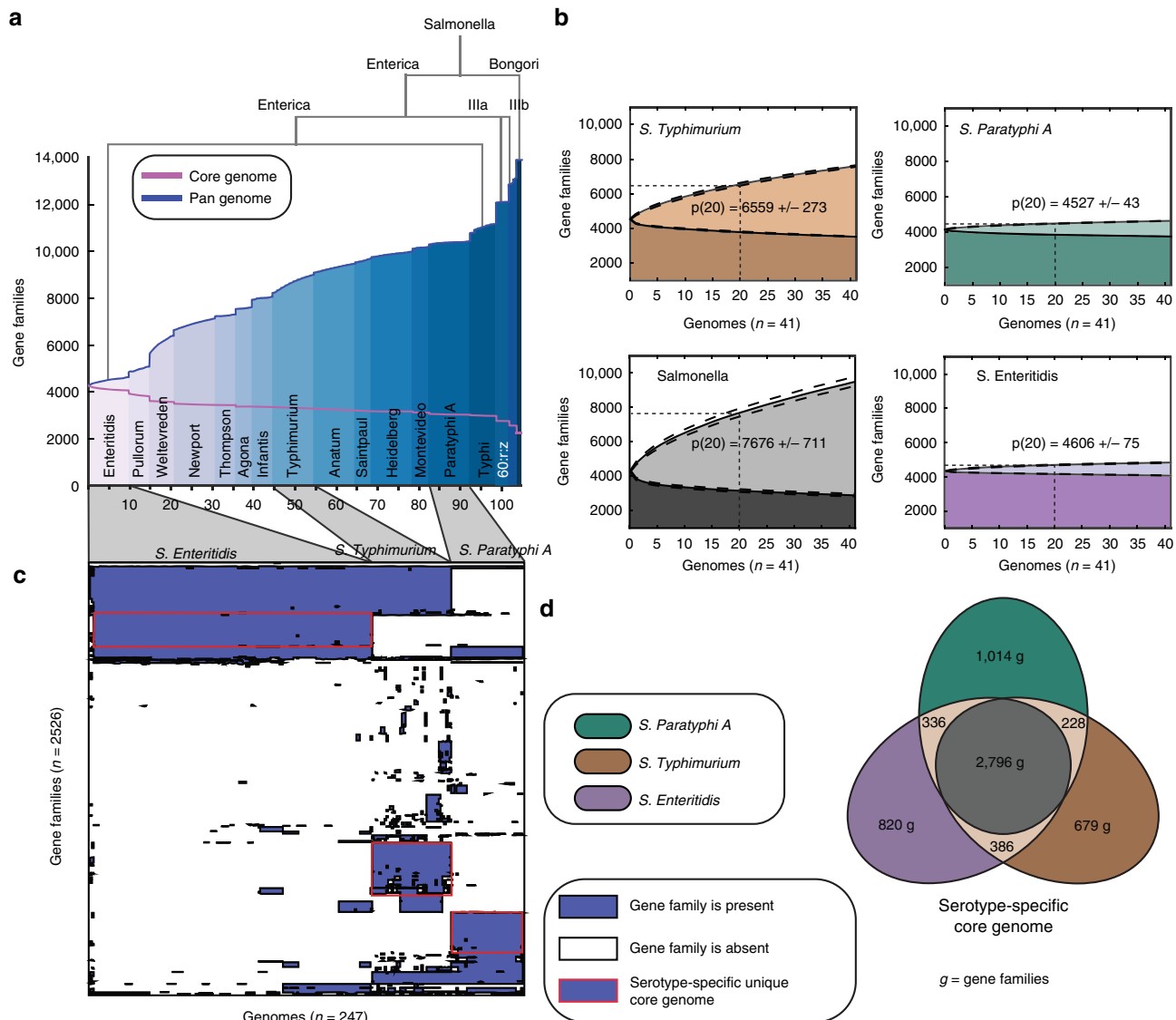

**Fig. 1** Core and pan-genomes of *Salmonella* serovars. **a** The *Salmonella* pan-genome was constructed for 104 out of 410 genomic sequences of *Salmonella*, including 2 species; *S. enterica* and *S. bongori*, 3 subspecies; *S. enterica* spp. *enterica, S. enterica* spp. *IIIA, S. enterica* spp. *IIIB*. For purposes of clarity, the serovars of *S. enterica* spp. *enterica* are represented in black when 3 or more corresponding genomes are available with a maximum of 10 randomly sampled genomes per serovar. See Supplementary Fig. 3 for the full *Salmonella* pan-genome. Note that branch widths do not correspond to phylogenetic distances. **b** Serovar-specific core and pan-genome curves for 41 genomic sequences of *Salmonella enterica* spp. *enterica*: Typhimurium (broad host range), Paratyphi A (host-restricted) and Enteritidis (broad host range) in comparison with 41 genomic sequences of randomly sampled *Salmonella* genomes. We show the average and standard deviation of the number of gene families at the 20th genomic addition (p(20)). **c** Cluster map of the accessory genome of Typhimurium, Paratyphi A, and Enteritidis. The gene families that are unique to a serovar and conserved across strains in that serovar are framed in red. **d** We identified the serovar-specific core gene families for serovars Paratyphi A, Typhimurium, and Enteritidis and plotted a venn diagram to represent the shared content. Paratyphi A has the highest number of core gene families, of which 1,014 are not part of the Typhimurium nor the Enteritidis core genome

**Characteristics of the *Salmonella* core and pan reactome.** To analyze the metabolic content of the accessory genome, we generated genome-scale network reconstructions (GEMs) of metabolism for each of the selected 410 *Salmonella* strains[15,16] (see Methods, Supplementary Data 3). Metabolic genes, metabolic reactions, and metabolites were compared across the strain-specific networks. We found that 1913 metabolic reactions are shared across all 410 strains (constituting the 'core metabolic reactome of *Salmonella*'), and 433 are present in some but not all strains (forming the 'accessory metabolic reactome') (Fig. 2, Supplementary Table 2).

The contents of the accessory reactome reveal that differences among *Salmonella* strains lie in part with their capability to

uptake and catabolize various nutrient sources. Metabolic processes involved in carbohydrate metabolism and inner membrane transport comprise a large percentage of the accessory reactome, 62 (14.3%) and 72 (16.6%) metabolic reactions and processes, respectively. These functional categories include alternate carbon metabolism, the glyoxylate cycle, periplasmic transport, and several catabolic pathways. Cell wall/membrane metabolism is also one of the least conserved subsystems and contributes to 14.7% of the total accessory metabolic reactome; a phenomenon that can be explained by the high O-antigen structural diversity across the 22 serogroups of *Salmonella* included in our dataset. The most highly conserved subsystems contained housekeeping functions that are common to all

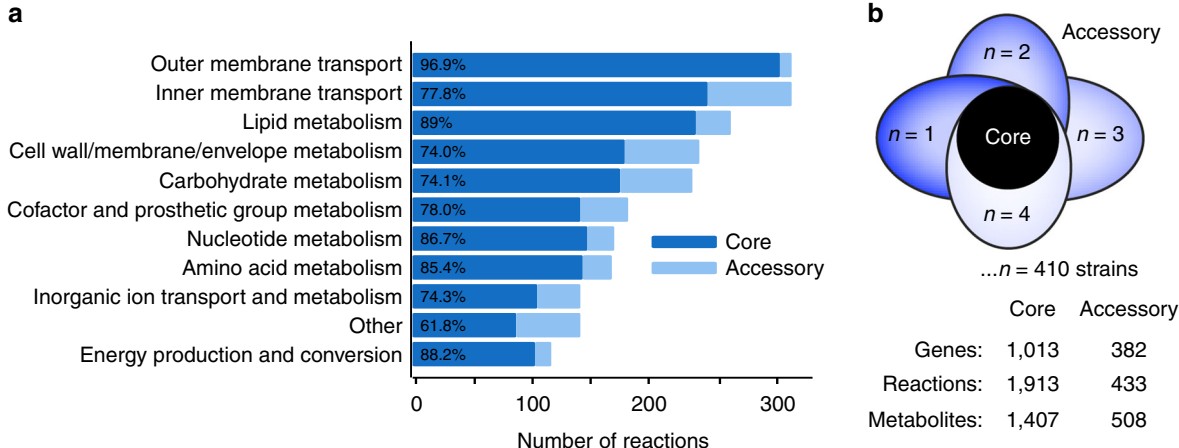

**Fig. 2** *Salmonella* pan reactome. **a** Reaction distribution per functional category in the 410 genome-scale models (GEMs) of *Salmonella* strains. The functional categories are defined by the clusters of orthologous groups ontology[64] and classify the metabolic reactions per subsystem. The reactome distribution across metabolic subsystems shows that inner membrane transport, carbohydrate and lipid metabolism constitute the largest portion of the pan reactome. The percentages shown represent the percentage of conserved metabolic processes in each functional category. **b** Number of core genes, reactions, and metabolites in 410 GEMs of *Salmonella*. A reaction, gene or metabolite is considered to be part of the core if present in all *Salmonella* GEMs

*Salmonella* strains. For example, 89.0% of the reactions and processes in lipid metabolism were part of the core metabolic reactome. Similarly, energy production and conversion (88.2%), nucleotide metabolism (86.7%), and amino acid metabolism (85.3%) were highly conserved.

**GEM-predicted growth capabilities differentiate serovars**. Because alternate carbon metabolism made up a large percentage of the accessory reactome (Fig. 2a), we hypothesized that these capabilities may reflect functional differences among strains in their capability to thrive in different nutrient environments. The conversion of metabolic network reconstructions into a mathematical framework allows for the computation of metabolic phenotypes based on the content of each reconstruction[19,20].

We leverage this functionality to simulate growth capabilities across all 410 *Salmonella* strain models on minimal media with 531 different growth-supporting carbon, nitrogen, phosphorous, and sulfur sources in aerobic and anaerobic conditions (see Methods, Supplementary Data 4). Major differences were observed in the predicted ability of different strains to catabolize myo-inositol (77.3% of strains incapable), D-Tagatose (69% of strains incapable) and D-Galactonate (20% of strains incapable) (Supplementary note 1, Supplementary Fig. 5). For example, all strains of serovars Typhi ($n = 6$), Paratyphi A ($g = 41$), Agona ($n = 4$), and Infantis ($n = 5$) were predicted to be incapable of utilizing D-galactonate as a sole carbon source while 98% of the strains of Typhimurium ($n = 46$) and all strains of Agona ($n = 4$), Infantis ($n = 5$), Thompson ($n = 5$), and Weltevreden ($n = 6$) could grow on myo-inositol as the sole carbon source. Additionally, while the full myo-inositol utilization operon is lost in 77.3% of strains, we observed partial gene loss across the tagatose utilization operon and the galactonate utilization operon. The genes of the tagatose operon involved in galactitol utilization were generally more conserved (Supplementary Note 2). Lactose is a major part of many human diets and galactitol is a byproduct of lactose catabolism (via glucose and galactose) suggesting that galactitol utilization could contribute to fitness of *Salmonella* strains.

We sought to examine catabolic capabilities that group strains of a serovar together and distinguish them from other strains. As a first step toward establishing such a classification schema, we compared predicted growth capabilities across the 8 serovars of

*Salmonella* that are represented with more than 5 GEMs each (Fig. 3a). We found that 5 out of the 8 serovars can be distinguished by their predicted capability to utilize various nutrient sources. For example, Paratyphi A strains are distinguished by their inability to utilize xanthosine 5′-phosphate while Weltevreden strains can utilize myo-inositol but cannot grow on D-tagatose or 2-aminoethylphosphonate as a sole carbon source. Similarly, 87% of Typhimurium strains can grow on D-tagatose, xanthosine 5′-phosphate, and myo-inositol as sole carbon sources. Trends were also observed across other serovars. For example, the four Montevideo strains were predicted to be incapable of utilizing allantoin and 2,3-diaminopropionate as the sole nitrogen and carbon source, and 3 out of 4 Saintpaul strains could not utilize 2,3-diaminopropionate as a sole nitrogen source. The prevalence of these losses suggests that they may have an effect on each serovar's preferred environmental niche.

**Catabolic capabilities reflect the host range of a strain**. We investigated whether we could use the computational characterization of bacterial metabolic networks to shed light on the content of an isolate's preferred ecological niche(s). We first subdivided the serovars into two groups: (1) the serovars that have been reported to thrive in a small subset of hosts (specialists); and (2) the serovars that are either known to colonize a variety of hosts or whose host specificity has not been explicitly demonstrated (generalists) (see Supplementary Data 1). There were 11 serovars that fell under the category of specialists, and 53 serovars that were classified as generalists. We hypothesized that if there are any links between the catabolic capabilities of a serovar and the host that it colonizes, those links would be more pronounced across the group of specialists.

We first constructed a cluster map from the computed growth phenotypes across all generalists. For clarity, we restricted our dataset to only show up to one GEM per serovar and its most variable catabolic capabilities across 19 media conditions (Fig. 3b). The full simulation outcomes can be found in Supplementary Data 4. We found that generalists mostly differ in their capability to utilize D-tagatose, myo-inositol, 2,3-diaminopropionate, allantoin, D-galactonate, and 2-aminoethylphosphonate.

Next, we constructed a decision tree to identify nutrient environments that could differentiate generalists versus specialists. We found that there was a general trend for host-restricted

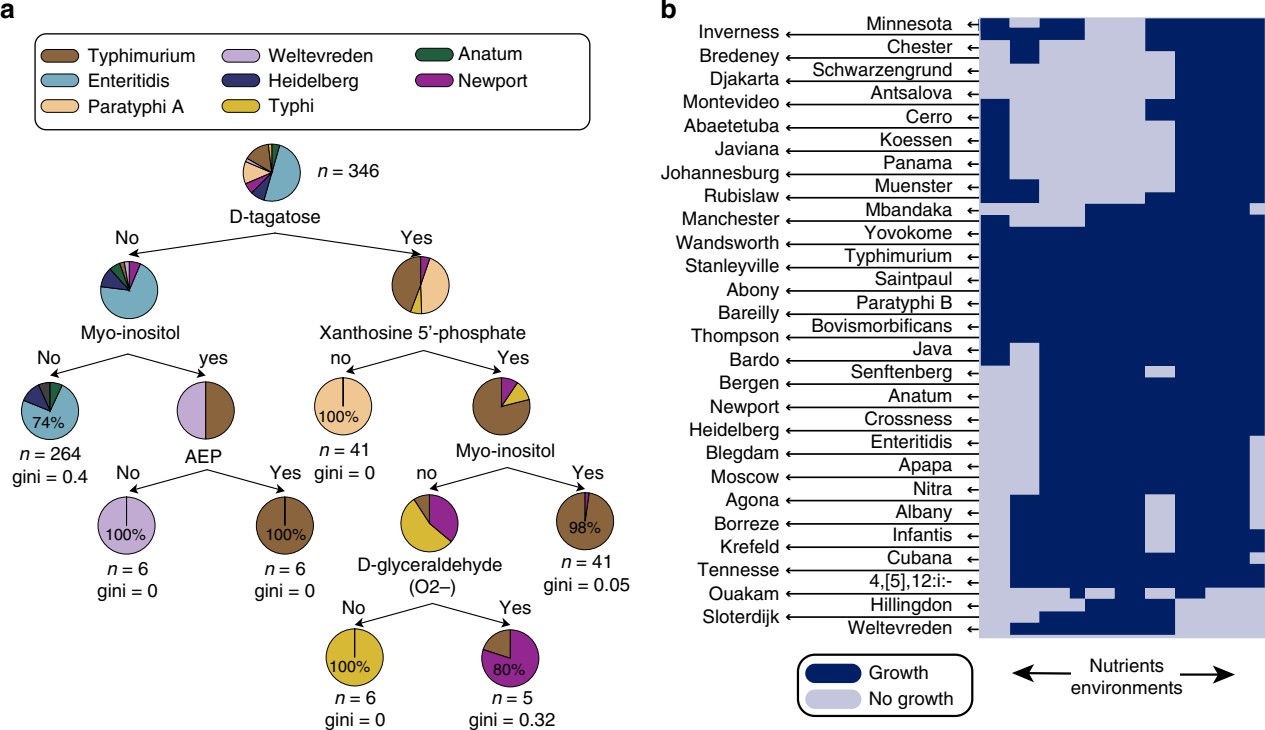

**Fig. 3** GEM-predicted catabolic capabilities across 53 non-host-specific serovars. **a** Eight serovars of *S. enterica* subsp. *enterica* can be classified based on their ability to catabolize six nutrients: D-tagatose, Xanthosine 5′-phosphate, *Myo*-inositol, D-glyceraldehyde (anaerobically) and Aminoethylphosphonate (AEP). For purposes of clarity, serovars were included in this classification when there were more than 5 strains in the dataset that represented it. The ability to catabolize a given nutrient always leads to the right while the inability to catabolize the listed nutrient leads to the left. Note that glycolate utilization was removed from the features used to build the decision tree (see Supplementary File 1). **b** The catabolic capabilities of the 410 GEMs across 323 nutrient sources in 532 media conditions were computed. Here the catabolic capabilities for 53 generalists across 19 simulated nutrient environments are shown (see Supplementary Fig. 4 for a detailed list). Each strain is randomly chosen as a representative of a serovar. Growth of a particular strain in a particular medium condition (represented by a dark blue color) demonstrates a positive GEM predicted nutrient utilization capability. The full dataset for catabolic capability predictions across all strains is found in Supplementary Data 4. Serovar names are listed along with an arrow pointing to the row it corresponds to in the cluster map

serovars to lose the capability to catabolize nutrients (Fig. 4a). In particular, strains of *S.* Enteritidis were predicted to be capable of catabolizing 4 more medium components than strains of *S.* Paratyphi A, namely formaldehyde, D-galactonate, xanthosine, and xanthosine-5-phosphate. Since *S.* Enteritidis is known to colonize a larger range of ecological niches, we hypothesize that strains of *S.* Enteritidis strains have more catabolic capabilities as a result of their lifestyle.

We next investigated whether catabolic capabilities can further serve to classify specialists by their specific niche (Fig. 4b). We found that host-restricted serovars known to colonize the same host lose similar catabolic capabilities. For example, the inability to grow in 15 nutrient environments (including L-asparagine, L-aspartate, L-malate, L-xylulose, and L-tartrate) distinguished serovars adapted to cold-blooded hosts from other specialists. Cold-blooded animal specialists (e.g., *S. bongori*, *S. enterica* subsp. *arizonae* and *S. enterica* subsp. *diarizonae*) shared the least number of metabolic capabilities, with successful growth for only an average of 485 out of 531 media conditions per strain. The extraintestinal human-restricted serovars (including *S.* Paratyphi A and *S.* Typhi, but not *S.* Paratyphi C) were the only specialists unable to degrade formaldehyde to formate and D-glyceraldehyde to glycerol.

All 41 Paratyphi A strains were additionally predicted to predominantly lack the metabolic capability to utilize xanthine or xanthosine-5′-phosphate as a sole carbon source (due to the absence of *xapAB*) with a total of 24 different nutrient conditions predicted to not support growth for at least one Paratyphi A

strain. In contrast, all 6 Typhi strains could not utilize L-idonate or 5-dehydro-D-gluconate as a sole carbon source (due to the absence of the four genes, including *idnDOTK*). Overall, there were a total of 24 different no-growth nutrient conditions predicted for at least one Paratyphi A strain. Choleraesuis strains are known to be swine-adapted and cause swine paratyphoid. They were found to lose the capability to utilize L-arginine as a sole carbon source under anaerobic conditions due to the absence of the succinylglutamic semialdehyde dehydrogenase (*astD*) and L-xylulose in both aerobic and anaerobic conditions due to the absence of three genes (*yiaMNO*). L-xylose is an upstream precursor of L-xylulose and is abundant in the hemicellulose walls of the cereals fed to pigs[21]. It was shown to be excreted in high proportions in the urine indicating very low levels of xylose absorption[21]. The loss of *yiaMNO* may have occurred because L-xylulose is not available in the swine extraintestinal environment.

**Nutrient utilization predictions demonstrate high-model accuracy.** We used an ensemble of 7 known metabolic traits to validate the reconstructed networks, including: growth on M9 minimal medium, fermentation of 11 carbon sources, production of hydrogen sulfide, growth on citrate as the sole carbon and energy source (with the exception of Typhi strains), capability to decarboxylate L-lysine, incapability to utilize lactose, and the presence of catalase. These tests constituted the basis for the exclusion of three genomic sequences from any subsequent analysis (PATRIC accession numbers 1412544.3, 1454644.3 and

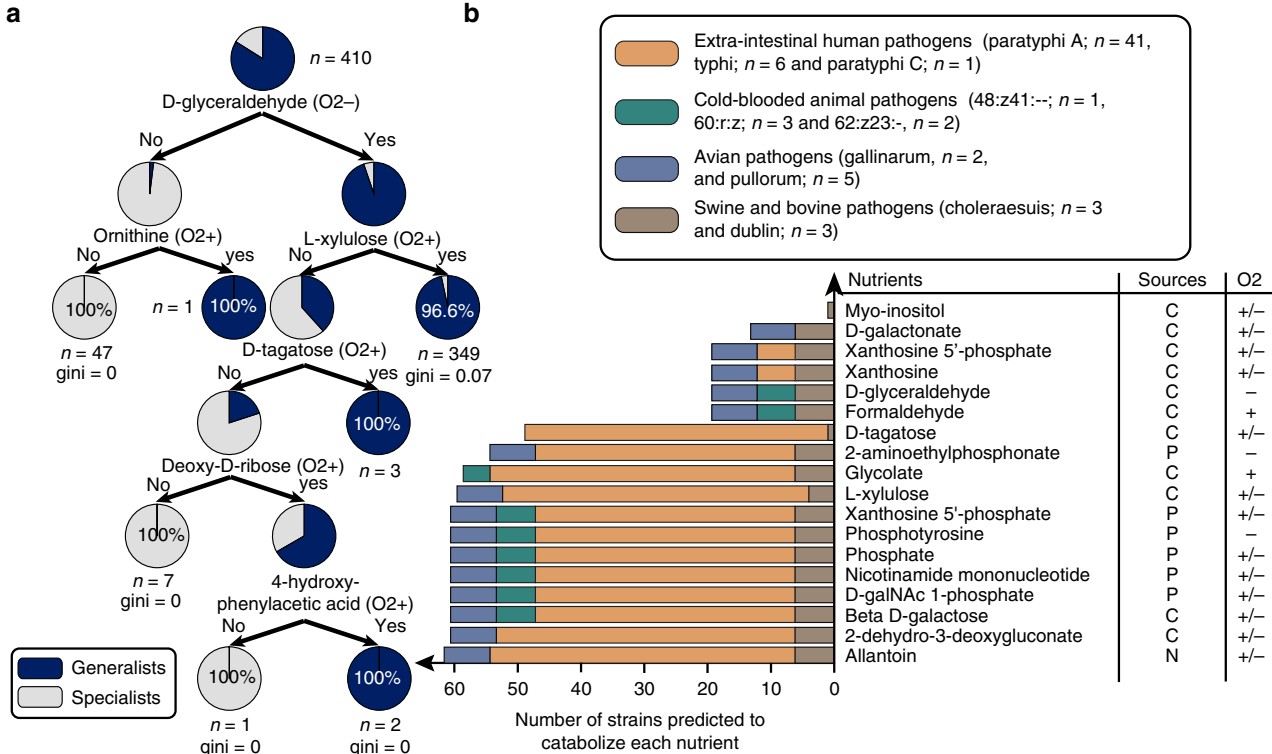

**Fig. 4** Catabolic capabilities across host-restricted serovars. **a** Decision tree depicting differential nutrient utilization capabilities in 410 GEMs classified as host restricted (specialists) or non-host restricted (generalists). Specialists are distinguished from generalists by a general trend toward losing catabolic capabilities. **b** Catabolic capabilities of 67 host-restricted strains on 18 nutrient conditions (see Supplementary Fig. 6 for heat map). The conservation of certain catabolic capabilities among serovars restricted to a common niche hints at the availability of certain nutrients in a niche and highlights the importance of that function for survival. The classification of each serovar by its known colonization niche is listed along with the number of strains per serovar included in the analysis, see Supplementary Data 1 for more details on the strains. The nutrient sources are classified as C for carbon, P for phosphate, and N for nitrogen source and their basis to support growth in aerobic (O2+) and anaerobic (O2−) conditions are shown

54388.108) as well as the identification of nutrient auxotrophies (see the next section, Methods).

We proceeded to validate the predicted strain-specific catabolic capabilities. We searched the literature for known catabolic capabilities of *Salmonella* strains and evaluated; (1) a dataset for 9 *S.* Typhi strains tested on a total of 190 carbon sources from Chai et al.[22] and (2) biolog phenotypic characterizations of 6 strains of *Salmonella* spanning 6 serovars (Typhimurium, Newport, Dublin, Heidelberg, Schwarzengrund, and Agona) on 59 different carbon sources from Fricke et al.[23] (Supplementary Data 5).

While the six strains of *S.* Typhi for which we have genomic sequences did not exactly match the nine strains tested in the first study, we assumed that the catabolic capabilities common across all experimentally tested strains would also be common among the strains used in this study. We first mapped 84 of the 190 carbon sources to their known catabolic pathways. Our GEMs correctly predicted 38 viable growth phenotypes and 19 no growth phenotypes across six strains (68%, fisher $p < 0.05$), but incorrectly predicted 27 no-growth phenotypes. We noticed that there are reports of hypothetically disrupted genes in *S.* Typhi known to be involved in the catabolic pathways for 7 of the 27 falsely predicted growth-supporting carbon sources, namely: citrate, L-glutamine, L-rhamnose, 1,2-propanediol, D-tagatose, ethanolamine and 4-hydroxyphenylacetic acid[24]. Pseudogene accumulation has been widely observed amongst Typhi strains[25] as well as other host-restricted strains of *Salmonella*[24,26]. We hypothesize that there could be additional disrupted genes involved in the utilization of some of the 20 remaining carbon sources. However, the identification of pseudogenes is beyond the scope of this paper. False positives may also indicate that the

pathways needed for growth under these conditions could be alternatively regulated[24,27].

The Fricke study also included matching genomic sequences for six strains that were tested for their capability to utilize a set of 59 carbon sources. Our corresponding GEMs correctly predicted 341 growth phenotypes, and 8 no growth phenotypes (98%, fisher $p < 0.05$). However, there were 5 failure cases across 3 strains in total (discussed in Supplementary Note 3). Overall these results demonstrate that the GEMs are of high-quality and that the predicted strain-specific metabolic traits can be used to generate hypotheses related to adaptation to specific hosts.

**GEMs enable investigation into the genetic basis of serovar-specific auxotrophies.** In addition to investigating growth-supporting nutrients, GEMs can also be used to examine the genetic basis of strain-specific auxotrophies. We found that GEMs for 32 out of 410 strains were unable to simulate the generation of at least one essential biomass constituent from glucose+M9 minimal medium without the addition of growth-supporting compounds to the in silico medium (Supplementary Data 6). We algorithmically filled these gaps in the metabolic network to examine the genetic basis for these potential auxotrophies (Fig. 5).

Based on this analysis, we found that 15 of the 159 *S.* Enteritidis strains exhibited an auxotrophy for tryptophan in silico. This observation is consistent with literature indicating that there are frequent occurrences of natural tryptophan auxotrophs across *Salmonella* serovars and other human pathogens[28–30]. A multiple sequence alignment of the *trp* operons across all genomes

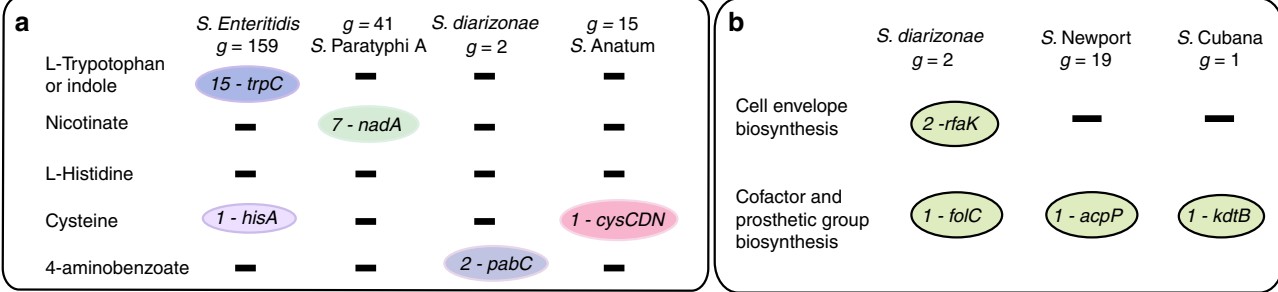

**Fig. 5** GEM-predicted auxotrophies and potential alternative metabolic pathways Serovars of *S. enterica* spp. *enterica* are highlighted: *S.* Paratyphi A, *S.* Anatum, *S.* Newport, and *S.* Cubana. **a** Nutrient auxotrophies are listed across serovars with the name of the missing gene that was found to cause the auxotrophy and the number of GEMs of that serovar affected. **b** Missing essential genes whose functions are likely carried out by an alternative unknown pathway in a serovar. *g* represents the total number of strains representing a serovar. For example there are 159 Enteritidis strains that were analyzed for auxotrophies. See Supplementary Fig. 7 for a multiple sequence alignment of the *trpC* locus

revealed a gap of 52 nucleotides or more at the *trpC* locus across the 15 predicted auxotrophs, which could indicate poor sequencing/assembly quality in this region (Supplementary Information). Beyond tryptophan, another predicted nutrient auxotrophy was found in seven of the 41 *S.* Paratyphi A GEMs that were predicted to be auxotrophic for nicotinate due to the absence of quinolinate synthase A (encoded for by *nadA*). Serovar Dublin isolates have been documented as being natural auxotrophs for niacin due to a missense mutation in *nadA*[28,31]. We observed that there was a trend for isolates of a serovar to be consistently associated with one or more auxotrophies (Fig. 5a). The majority of in silico-predicted auxotrophies were amino acid-based: L-tryptophan (15 strains), cysteine (1 strains), L-histidine (3 strains), L-threonine (1 strain) and *beta*-alanine (1 strain). There is ample literature evidence of amino acid auxotrophies among *Salmonella* isolates and several genera of bacteria[14,32], and it has been shown that these amino acids can be scavenged from the environment and used as a main carbon source[33].

Our dataset also included two GEMs for *S. enterica* spp. *arizonae* IIIa. There were two metabolic capabilities essential in *S.* Typhimurium GEMs that could not be extrapolated to these GEMs solely based on sequence homology: dihydrofolate synthesis (deletion in *folC*) and heptose transfer IV involved in lipid A core oligosaccharide biosynthesis (deletion in *rfaK*). *S. enterica* spp. *arizonae* is known to express a different core oligosaccharide structure with a glucose residue at the terminal end instead of N-acetyl-glucosamine[34] (Fig. 5b). Interestingly, both GEMs were missing genes involved in the metabolic pathway for de novo folate biosynthesis—a function that is essential for survival and a target of antifolate antibiotics[35]—and were thus *p*-aminobenzoate auxotrophs. Alternative pathways may exist to compensate for this metabolic requirement

**Catabolic capabilities impact fitness across hosts.** We next sought to determine which nutrient utilization pathways affect the fitness of *Salmonella* strains in their natural microenvironments and host ranges. In previous studies, 10,000 Tn5 mutants of *S.* Typhimurium strain SL1344 were tested for fitness in intravenous infection of BALB/c mice[36], and 9792 Tn5 mutants of *S.* Typhimurium strain ST4/74 were tested for fitness in oral infection of pigs, cattle, and chickens[37]. Using these datasets, we set out to identify which catabolic pathways uniquely lost by some serovars conferred an advantage in one environment but not the other. To link genes in the reconstructed *Salmonella* pan-reactome to the conferred catabolic capabilities, we searched for gene essentiality in all simulated nutrient environments. Here we defined conditionally essential genes (CEG) as those

genes found to be essential in one of the nutrient conditions but not in aerobic M9+glucose minimal medium. Of the 531 nutrient environments, 242 were anaerobic and 289 were aerobic. We identified a total of 242 predicted CEGs, of which 195 and 217 were essential in at least one aerobic and at least one anaerobic nutrient condition, respectively, with some genes being essential in both (Fig. 6a, Supplementary Note 4, Supplementary Data 7).

Of the 242 predicted CEGs, several were shown to contribute to fitness in colonizing the mouse spleen (23 CEGs), as well as the intestine of cattle (53 CEGs), pigs (40 CEGs), and chickens (42 CEGs). Only nine CEGs contributed to fitness in all hosts, whereas 30 CEGs contributed to fitness in cattle, pigs, and chickens (Fig. 6b). The nine CEGs shared across hosts included genes that were essential in many of the in silico environments (mCEGs) such as *atpAD* (two subunits of ATP synthase), *galE* (UDP-glucose 4-epimerase), *gor* (glutathione oxidoreductase), and *ptsI* (a subunit of the phosphoenolpyruvate-protein phosphotransferase involved in non-specific carbon transport). However, CEGs specific to single in silico environments (sCEGs) were also essential across all hosts, including *mtlD* (mannose-6-phosphate isomerase), *sucC* (succinyl-CoA synthetase), *yiaN* (subunit of the L-xylulose transport complex), and *manA* (mannose-6-phosphate isomerase). The sCEGs represent genes required for growth on specific carbon substrates, for example: *manA* is essential when D-mannose or D-mannose-6-phosphate serves as the sole carbon source; *mtlD* is essential when D-mannitol is the sole carbon source; and *yiaN* is essential when L-xylulose serves as the only source of carbon.

Of the 30 CEGs with unique contributions to fitness across the intestinal infection of pigs, chickens, and cattle, 21 do not contribute to fitness during splenic infection of mice (Table 1), including 17 sCEGs and four mCEGs. In this set, 13 of the 17 sCEGs are conditionally essential for the utilization of D-tagatose, allantoin, deoxy-D-ribose, L-tartrate, D-xylose, D-xylulose, L-idonate, allantoin, L-arginine, 2,3-diaminopropionate, D-glyceraldehyde, and formaldehyde, all of which were predicted to be incapable of supporting growth of at least one host-associated strain.

Our predictions indicate that the serovars adapted to cold-blooded hosts could not grow in seven of these media conditions, including: D-tagatose, D-xylose, deoxy-D-ribose, L-xylulose, L-tartrate, 2,3-diaminopropionate and 4-aminobutanoate. For example, all three *S. bongori* strains were missing *deoP* and *deoK*, both of which are essential for the utilization of deoxy-D-ribose, and both of which were also shown to contribute to fitness in cattle, chickens, and pigs, but not in mice (Fig. 6c). A possible explanation for these altered growth characteristics

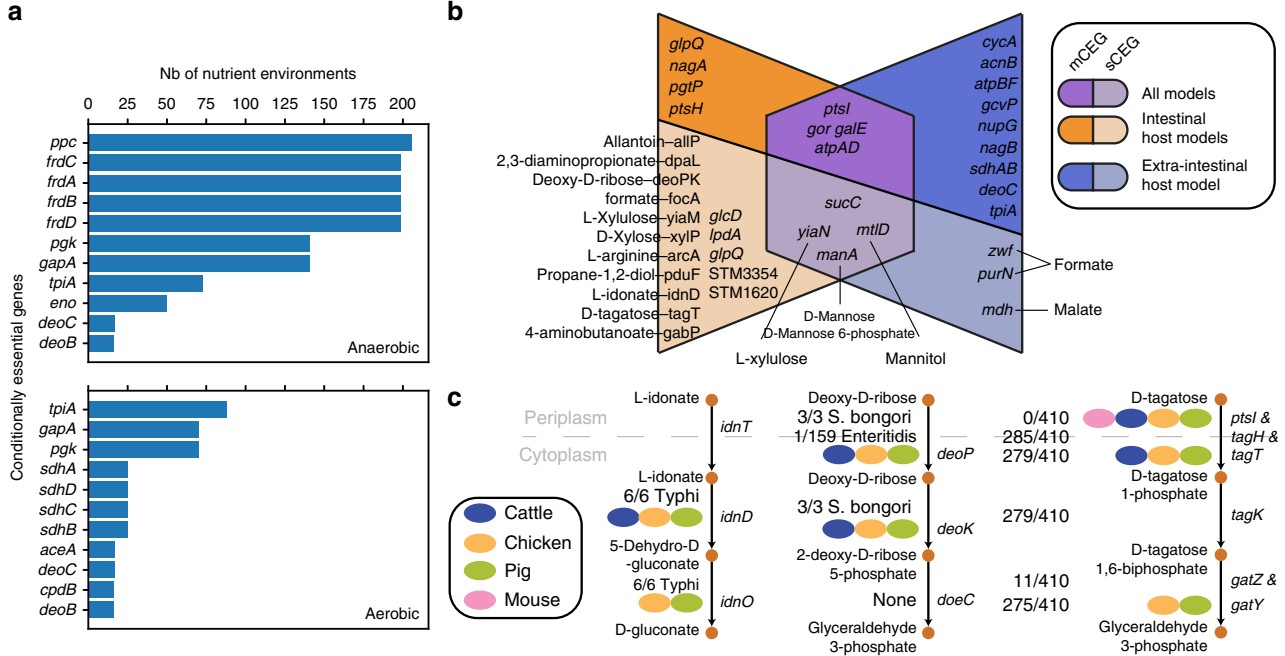

**Fig. 6** Conditionally essential genes (CEGs) and corresponding mutant fitness in diverse hosts. **a** We searched for CEGs across 531 nutrient environment conditions and found a total of 20 involved in central metabolism. Of the 531 nutrient environments, 242 were anaerobic and 289 were aerobic. We plot here the 10 most frequent CEGs in aerobic and anaerobic conditions. **b** We selected for CEGs whose corresponding mutant was found to contribute to fitness in at least one host. We then plotted a venn diagram of CEGs that were observed to be important for fitness in intraintestinal versus extraintestinal hosts. We subdivided CEGs into those that were predicted to be essential in 5 or more nutrient environments (mCEGs) and those predicted to be essential in less than 5 nutrient environments (sCEGs). **c** We identified the occurrence of sCEGs that were seen to contribute to fitness across strains of *Salmonella*. We highlight here three catabolic pathways featuring the selected sCEGs and the number of *Salmonella* strains that do not carry the CEGs in their genome. The genes are placed next to the metabolic process that they are involved in. The fitness contributions of an sCEG to hosts are highlighted (ovals indicate that a significantly affected fitness was measured in this host)

| Table. 1 Conditionally essential genes observed to affect fitness in cattle, chicken, and pig but not mouse spleen, and their calculated TRADIS fitness score | | |
|---|---|---|
| **Locus tag** | **Encoded reactions** | **Nutrient conditions** |
| STM1002 | 2,3-diaminopropionate ammonia lyase | (N-O2+/−, C-O2+) 2,3-diaminopropionate |
| STM4484 (*idnD*) | ʟ-idonate 5-dehydrogenase | (C-O2+/−) ʟ-Idonate |
| STM0042 (*xylP*) | ᴅ-xylose transport in via proton symport | (C-O2+/−) ᴅ-Xylose |
| STM3354 | ʟ(+)-tartrate dehydratase | (C-O2+/−) ʟ-tartrate |
| STM3255 (*tagT*) | ᴅ Tagatose transport via PEPPyr PTS | (C-O2+/−) ᴅ Tagatose |
| STM3792 (*deoP*) | Deoxy D ribose transport via proton symport | (C-O2+/−) Deoxy D Ribose |
| STM3793 (*deoK*) | Ribokinase and deoxyribokinase | (C-O2+/−) Deoxy D Ribose |
| STM4467 | Arginine deiminase | (C-O2-) ʟ-Arginine |
| STM2793 (*gabP*) | 4-aminobutyrate transport in via proton symport | (C-O2+, N-O2+/−) 4-Aminobutanoate |
| STM1627 | Formaldehyde dehydrogenase and Glycerol dehydrogenase | (C-O2+) Formaldehyde and (C-O2-) ᴅ-Glyceraldehyde |
| STM0522 (*allP*) | Allantoin transport in via proton symport | (N-O2+/−, C-O2+/−) allantoin |
| STM0974 (*focA*) | Formate transport via proton symport | (C-O2+) Formate AND (C-O2-) formaldehyde |
| STM3671 (*yiaM*) | ʟ-xylulose transport in via proton symport | (C-O2+/−) ʟ-Xylulose |
| STM2282 (*glpQ*) | Glycerophosphodiester phosphodiesterase | (C-O2+/−, P-O2+/−) Sn-Glycero-3-phosphocholine |
| STM2037 (*pduF*) | (R)-Propane-1,2-diol facilitated transport | (C-O2+) (R)-Propane-1,2-diol |
| STM1620 | Glycolate oxidase | (C-O2+) Glycolate |
| STM0154 (*lpdA*) | Pyruvate dehydrogenase and 2-Oxogluterate dehydrogenase | (C-O2-) DCMP and Deoxyuridine and DUMP and Deoxycytidine |

(46 no-growth media conditions on average) is that the microenvironments where these cold-blooded host-associated strains normally reside markedly differs from the intestinal environment of farm animals. Similarly, all strains of Typhi were predicted to lack the capability to utilize ʟ-idonate due to the absence of *idnD* and *idnO*, which correlates with Typhi's inability to grow on ʟ-idonate as a sole carbon source[22]. These genes demonstrated fitness defects during intestinal infection of the farm animals assayed, but not in spleen of infected mice.

ʟ-ascorbate (vitamin C) is an essential nutrient in the human diet, and ʟ-idonate is an intermediate product of ʟ-ascorbate catabolism which has been shown to decay spontaneously in vitro[38,39]. Taken together, these findings suggest that ʟ-idonate is an available nutrient source in the gut[38,39]. The fact that *S.* Typhi is known to colonize the extraintestinal environment and is specific to humans suggests that the loss of this capability comes as an adaptation event in which genes that do not contribute to fitness have been lost.

## Discussion

In this study, we built serovar-specific pan-genomes and reconstructed strain-specific genome-scale metabolic models from the genomes of 410 *Salmonella* strains. Using the strain-specific GEMs, we: (1) compared and contrasted core and pan metabolic capabilities within the *Salmonella* genus; (2) determined differences among serovars in growth phenotypes on over 530 different media; (3) explored the genetic basis of underlying strain-specific auxotrophies; (4) identified candidate catabolic pathways that contribute to the fitness of *Salmonella* strains in diverse microenvironments; and (5) examined the occurrence of those catabolic pathways across different *Salmonella* strains.

The pan-genome analyses revealed that the three main *Salmonella* serovars have pan-genome sizes that do not seem to reflect their host range. The total number of gene families in a random sample of 20 genomes of Enteritidis (a generalist) was similar to that of Paratyphi A (a specialist) but much lower than that of Typhimurium (a generalist). Since the ability to colonize multiple niches does not seem to affect the pan-genome size[40], we asked whether specific classes of genes in the pan-genome reflected this ability. A comparison of the gene family content across three serovars revealed that a specific repertoire characterizes each serovar, and that while certain gene families were part of the *Salmonella* accessory genome, they appear in a given serovar's core genome.

While pan-genome analyses provide useful insights into the genetic variability amongst strains and serovars, strain-specific genome-scale models allow for explicit prediction of phenotypes. GEMs are mathematically structured knowledge bases that have been validated against experimental data and have demonstrated high accuracy for growth phenotype predictions[11,12]. The majority of *Salmonella*'s core reactome consisted of lipid metabolism as well as energy production and conversion. By contrast, the pan-reactome revealed that differentiating features across strains lay in their cell wall composition and their unique capabilities to transport and catabolize specific nutrients. The cell wall composition is a trait that has been exploited for the characterization of serovars. Therefore, we asked whether catabolic capabilities can also serve to differentiate serovars.

To date, relatively few metabolic traits have been identified that distinguish *Salmonella* serovars. For instance, *S.* Typhi can be distinguished from other *Salmonella* isolates, in part because it is citrate negative[28], whereas L-tartrate utilization distinguishes extraintestinal and gastrointestinal strains of *S.* Paratyphi B[41]. We set out to identify additional differentiating catabolic capabilities using the reconstructed strain-specific GEMs. Model-simulated growth on different nutrient sources demonstrated that strains of each serovar clustered together. Leveraging this finding, we then built a decision tree that distinguished 5 serovars based on their model-predicted catabolic capabilities. There was a tendency among host specialists to lose catabolic capabilities, while serovars specific to the same niche tended to share similar catabolic profiles. When we compared a total of 858 predicted growth phenotypes with experimental observations, we obtained an overall 83.1% agreement, demonstrating high-model accuracy. The discrepancies likely indicate the presence of pseudogenes.

We proceeded to ask whether the observed gene losses across specialists are a result of the composition of their microenvironment. We first identified candidate nutrient catabolic pathways that contribute to the fitness of *Salmonella* strains across hosts. We mapped nutrient conditions to the corresponding predicted conditionally essential genes (CEGs), and found that the fitness of 9 *Salmonella* mutants (including mutants in *gor*, glutathione oxidoreductase) were shown to be affected in all four hosts. Glutathione oxidoreductase was predicted to be conditionally essential for the utilization of trithionate, thiosulfate or tetrathionate as the sole sulfur source[42], and tetrathionate is known to confer an advantage to *Salmonella* strains which utilize it as a terminal electron acceptor in the inflamed gut[43].

We then asked whether the differential fitness conferred by catabolic genes revealed important variations in the nutrient composition across hosts. Indeed, a total of 21 catabolic pathways uniquely contributed to *Salmonella* fitness during intestinal infection of pigs, cattle, and chickens, including those for the utilization of D-tagatose, L-xylulose, D-xylose, deoxy-D-ribose, L-idonate, D-glyceraldehyde, and allantoin. These metabolites form part of the host's diet and/or have been observed in the intestinal environment[21,44,45]. Additionally, we observed that the corresponding catabolic capabilities were missing across strains adapted to either cold-blooded hosts, to swine, or to humans, providing evidence of an isolate's genotype being influenced by the environment it evolves in. For example, Typhi strains lost the capability to utilize L-idonate. Indeed, the differential fitness conferred by genes involved in catabolic pathways across hosts possibly reflect compositional differences in the intestinal versus extraintestinal milieu and may also reflect differences in pathogenicity[46].

In addition to identifying unique growth capabilities, the GEMs also predicted strain and serovar-specific auxotrophies. Auxotrophies can indicate cases of directed evolution to a new host, where ancestral traits that interfere with virulence are lost. The ability of GEMs to predict serovar-specific auxotrophies makes them a powerful tool to elucidate the evolutionary trajectory of various serovars. Using the GEMs we identified two predominant auxotrophies, including a niacin auxotrophy in seven of the *S.* Paratyphi A strains. Intriguingly, this auxotrophy has been shown to enhance the virulence of *Shigella flexneri* strains in humans[47].

Altogether, our study demonstrates that strain-specific models of *Salmonella* metabolism can be used to systematically identify a serovar's unique metabolic capabilities. These capabilities may affect host range and the preferred environmental niche of a given serovar. Moreover, these results represent a step toward the definition of a bacterial serovar based on GEM-predicted metabolic capabilities. In addition to this fundamental advance, nutrient utilization characteristics provide a basis for understanding strain and serovar-specific pathogenesis. Ultimately, this understanding could be leveraged to formulate strain- and serovar-specific drug development and therapeutic approaches.

## Methods

***Salmonella* genome sequence selection**. A collection of 439 closed genomic sequences were downloaded from public databases[15,16]. These genomes were selected when they were annotated as "complete" on PATRIC and "chromosome" or "complete genome" on NCBI[48]. None of the records contained contigs, i.e., the sequence was continuous. While some genomic sequences were complemented by the respective plasmid(s), others were not. Thus, in order to normalize the data, all plasmidic sequences were excluded from further analyses. Draft genome-scale models were initially built for all sequences. In silico validation tests, described in the section titled *GEMs validations* resulted in the exclusion of 29 genomic sequences and the identification of an additional threshold for the quality of an assembly in the context of constraint-based modeling. In short, a genomic sequence was excluded when it contained more than 70 regions of 30 contiguous unassigned nucleotide bases (Supplementary Data 1). We observed that such regions resulted in ORF disruption and erroneous ORF calling. A total of 410 genomic sequences passed that threshold of which 72 were drawn from the NCBI database[15] and 338 were drawn from the PATRIC database[16]. To remove any inconsistencies in ORF calling annotations across annotation platforms, all 410 genomic sequences were re-annotated using Prokka[49]. Pseudogenes were not manually annotated beyond this point to avoid increasing the false positive rate of gene absence.

***Salmonella* pan-genome construction**. Sequence homology was used to cluster genes into gene families using a clustering tool namely, CD-Hit[50]. Genes from all genomes were extracted and fed into CD-Hit. The sequence identity threshold was set at 0.9 and the word length was set to the default of 5. A gene family ID was then assigned for each gene. The pan-genome consisted of the collection of all gene

families. Gene family IDs were used to identify the shared gene families across genomes.

**Multi-locus sequence alignment housekeeping genes**. We downloaded a database containing variants of *Salmonella* housekeeping genes[51], including *aroC*, *dnaN*, *hemD*, *hisD*, *purE*, *sucA*, and *thrA*. We proceeded to search for homology of these genes across all genomic sequences by using BLAST. We selected a lower threshold of 80% identity and an upper threshold of $10^{-5}$ e-value. We then concatenated the 7 genes (in the same order) and aligned the sequences using MUSCLE[52]. We subsequently computed the phylogenetic distance matrix using Distmat[53].

**Salmonella serovar-specific pan-genome comparison**. All pan and core genome curves were constructed using the set of gene families obtained in the previous steps. Pan-genome curves were built by drawing from a set of genomic sequences one at a time without replacement and summing the number of novel gene families encountered at each draw. Core curves were built by drawing from a set of genomic sequences one at a time without replacement and subtracting the number of gene families not encountered in the new draw. The core and pan-genome size were determined at each draw. Traditionally, the genomic sequences are sampled (i.e., their order is mixed at random) and a pan and core genome curve are drawn for each sample.

Here, the available set of 410 genomes was subdivided into 4 subsets consisting of: (1) 41 sequences for strains of Paratyphi A, (2) 46 sequences for strains of Typhimurium, (3) 159 sequences for strains of Enteritidis, and (4) 410 sequences for all strains of *Salmonella*. Serovars Paratyphi A, Typhimurium, and Enteritidis were chosen because there were more than 40 available genomic sequences available for each. From each subset, we randomly sampled 41 sequences to obtain 1000 genomic permutations. In this way, we reconstructed four sets of pan and core genome curves (Fig. 1b). Fitting the curves with Heap's law resulted in large variance in the fitted parameters (Supplementary Note 5, Supplementary Fig. 3 and 8). Instead, we chose to report the average and standard deviation of the number of gene families found at a selected genomic addition across all pan-genome curves. We selected the genomic addition after half of the dataset had been included in the pan-genome analysis so as to obtain a standard deviation that is reflective of the population. Since we sampled 41 sequences, we reported the number of gene families encountered at the 20th genomic addition.

Because we noticed that the core genome size was different for each subset, we applied the unpaired Student's *t*-test to assess whether there was a difference in the number of shared gene families between two strains of the same serovar and two strains of different serovars. For that purpose, two distributions were generated. Two genomes were randomly sampled from all 410 genomes 500,000 times. For each pair of genomes, the number of shared gene families was computed. When the two sampled strains belonged to the same serovar, the number of shared gene families was added to the first distribution, otherwise it was added to the second distribution. As a result, the first distribution contained 15,154 computed numbers and the second distribution contained 67,061. The normality of the two distributions was confirmed ($p < 0.001$) using the "stats.mstats.normaltest" command from the scipy toolkit. The unpaired Student's *t*-test was then applied to determine whether the two distributions were significantly different using the "stats.ttest_ind" command from the scipy toolkit. Since the *p*-value was <0.001, we concluded that they were.

To represent the fact that two strains of a similar serovar shared significantly more gene families than two strains of different serovars, we modified the traditional workflow that computes core and pan-genome curves. We started by grouping strains of the same serovar together and only sampled those strains together. Thus, genomic sequences from the first serovar were sampled 100 times and the core and pan-genome values were averaged after each genomic addition. The genomic sequences from the second serovar were subsequently introduced and core and pan-genome values were computed. After the addition of the last serovar to the pan-genome, the core and pan-genome curves were finally plotted (Fig. 1a and Supplementary Fig. 3).

To examine the pan-genome in more detail, we constructed a cluster map for the gene families across 247 genomes of serovars Typhimurium, Paratyphi A, and Enteritidis using the *clustermap* command from the seaborn package (Fig. 1c). We excluded the gene families that are shared by all genomic sequences. As a result, we introduced a new term; "the serovar-specific core genome", which represents the set of gene families that are shared across all strains of a serovar. Accordingly, the core genome of a species is the set of core genomes of all the serovars of that species. The venn diagram in Fig. 1d displays the number of shared gene families across serovar-specific core genomes for serovars Typhimurium, Enteritidis, and Paratyphi A.

**Reconstruction of a consensus Typhimurium str. LT2 GEM**. We used the genome scale metabolic reconstruction (GEM) for *S. enterica* ser. *Typhimurium str. LT2* (STM.v1.0) as a starting point for our reconstruction efforts[13,54]. A GEM is a curated structured knowledge base that contains all of the biochemical transformation occuring in a cell along with a mapping of the gene encoding them[55]. The starting GEM contained 2545 reactions, 1271 genes, 1802 metabolites, uptake rates

for in vitro M9 minimal medium (see Troubleshooting and gap-filling for a detailed definition) and the corresponding biomass reaction. We updated this network with curation efforts performed by another group[14]. Additions included a new in vivo biomass reaction with experimental validation, 16 new genes and 48 new reactions spanning various subsystems, including 20 reactions involved in transport and exchange, 15 reactions involved in cell envelope biosynthesis and 4 reactions involved in cofactor and prosthetic group biosynthesis. We also updated the *Salmonella* core oligosaccharide biosynthesis pathway by removing the rhamnosyltransferase reaction because the core oligosaccharide structure in *Salmonella* strains does not contain a rhamnose residue. Because different strains of *Salmonella* possess a wide range of O-antigens and addition of synthesis capabilities for these was outside the scope of this project, we removed O-antigens from the in silico *Salmonella* biomass objective function. O-antigens form the outer part of the LPS molecule—which are glycolipids that are embedded in the external membrane of Gram-negative bacteria. They are variably produced throughout a bacterial lifetime, vary in structure across serogroups and are not always necessary for the strain's survival[56]. They thus do not strictly fit within the definition of a biomass objective function[57].

**Functional annotation of orthologous proteins via Uniprot**. The annotated and manually reviewed *Salmonella* protein sequences were queried from Swiss-Prot[58]. All 1287 manually curated proteins included in the *Salmonella* GEM were then blasted against the set of Swiss-prot proteins. The best bi-directional BLAST hits (BBH)[59] were selected and detailed biochemical assignments were extrapolated for all orthologs/gene variants of a cluster. A BBH pair is defined when two genes are best BLAST hits of each other with a minimum percent identity threshold of 80% and a maximum e-value of 0.01. As a result, 2131 *Salmonella* protein sequences in Swiss-Prot were annotated with biochemical reactions using BiGG standards[60] and added to the pan-STM.v1.1 reconstruction.

**Pan-STM.v1.1 expansion using the BiGG database**. Metabolic networks and proteins are often shared across closely related species. We thus queried 55 genome-scale reconstructions of species closely related to *Salmonella* from BiGG to build a metabolic reconstruction database. The species included *E. coli* (46 GEMs)[11,61], *Shigella* (8 GEMs)[11,61], and *K. pneumoniae* (1 GEM)[62] (Supplementary Data 2).

We used BLAST to map all *Salmonella* gene families against the BiGG database built above and assigned candidate gene reaction rules based on BBH pairs. When the function was already represented in the pan-STM.v1.1, but the sequence was novel, the protein was considered to be an ortholog. When the function was novel, it was checked against the literature for evidence of its presence in *Salmonella* and assigned a confidence score that complies with the standards set by Thiele et al.[55]. Functions with a confidence score of more than 0 were added to the reconstruction. As a result the pan-STM.v1.1 was expanded with 124 new reactions, 119 new genes, 96 new metabolites, and 53 new orthologs. These reactions spanned several subsystems notably the inner and outer membrane transport, amino acid metabolism, membrane lipid metabolism, antibiotic resistance, nucleotide salvage pathway, pentose phosphate pathway, cofactor, and prosthetic group metabolism. All new reactions, metabolites, and genes were standardized using BiGG abbreviations. The pan-STM.v1.1 reconstruction included 2695 reactions, 1395 genes, 2131 gene orthologs, 1935 metabolites, and biomass objective functions representing in vivo and in vitro conditions (Supplementary Data 2). We proceeded to run a homology search using bi-directional best blast hits of all of the genomic sequences included in the dataset against the pan-STM.v1.1 annotated genes (Supplementary Data 3).

**Medium definition and biomass objective function**. A biomass reaction represents the drain of precursors and macromolecular building blocks for bacterial growth in a certain environment (i.e lipid, glycogen, lipopolysaccharides, amino acids, and nucleotides). Quantifications are based on the experimental determination of the relative fractions of the precursors in a culture[57]. As such, the biomass reaction highly depends on the bacterial environmental niche. There have been two such formulations in *Salmonella* for growth in M9+glucose minimal medium (in vitro) and mouse spleen tissue (in vivo) both of which have been experimentally verified. The M9+glucose minimal medium definition includes less restrictive uptake rates and fewer nutrient sources. It is simulated with unlimited uptake rates for Calcium, Chloride, $CO_2$, $Co^{2+}$, $Cu^{2+}$, $Fe^{2+}$, $Fe^{3+}$, Potassium, Magnesium, $Mn^{2+}$, Molybdate, Sodium, Ammonium, Phosphate, Sulfate, Tungstate, and Zinc and limited uptake rates for 2,3 Di-hydroxybenzoate, $O_2$, $H^+$, $H_2O$, D-Glucose, and Cob (1)alamin (Supplementary Data 3). By convention, exchanges are negative when the flux is directed from the extracellular compartment to the periplasmic compartment. For the purpose of predicting growth, we ran flux balance analysis (FBA) using the COBRApy package version 0.13.2 to simulate for an optimal flux state with the in vitro biomass reaction set as the objective function. Flux balance analysis is a method to analyze the flow of metabolites through a metabolic network[19].

**GEM validations**. The pan reactome of *Salmonella* served as a scaffold for building the strain-specific reconstructions. All genomic sequences were blasted against the

annotated genes in the pan-STM.v1.1 reconstruction. In each genome, gene presence/absence was determined on the basis of sequence homology between Prokka-predicted coding DNA sequences and the curated genes. We considered genes to be homologous when they were BBH pairs. The result of this step led to the creation of 410 strain-specific GEMs. We ran several safety checks across the reconstructions to assess their validity including: (1) FBA (to simulate growth); (2) fermentation of various carbohydrates (Lactose, glucose L-arabinose, maltose, D-mannitol, D-mannose, L-rhamnose, D-sorbitol, trehalose, D-xylose and dulcitol); (3) growth on citrate as the sole carbon source; (4) carboxylation of lysine, and; (5) production of hydrogen sulfite. These checks served to verify that all GEMs could recapitulate some of the known catabolic/metabolic features of Salmonella[63]. A GEM cannot simulate growth if one of the biomass precursors cannot be produced, which occurs when there is a gap in one of the essential anabolic pathways. We assumed that a maximal growth of $<0.001\,h^{-1}$ meant that no growth could be achieved.

We tested the capability to ferment various carbohydrates by: (1) setting the lower bound for oxygen and glucose exchange to 0 mmol/gDW/h, (2) iterating through each of the carbon sources by setting the lower bound for the corresponding reaction at −50 mmol/gDW/h and, (3) optimizing for flux through the biomass function at each iteration. Growth on citrate as the only carbon source was simulated similarly but with the lower bound for oxygen set to −20 mmol/gDW/h (its default amount). Production of hydrogen sulfite was simulated by the temporary addition of a demand reaction for hydrogen sulfite with the objective coefficient set to 1. The biomass objective coefficient was set to 0 and FBA was employed to simulate flux. When a test failed, i.e. growth could not be achieved, the model was flagged for further investigation. Other identification tests that characterize Salmonella by its inability to produce a certain enzyme or to catabolize certain substrates were not taken into consideration here because they are already embedded in the curated metabolic reconstructions.

**Troubleshooting and gap-filling**. All previously flagged models were considered and distributed into four classes: (1) the flag was raised because of the presence of unknown alternative pathways, (2) the modeled strain is a potential auxotroph, and (3) there was an assembly and/or annotation error.

We first identified 160 essential genes for biomass production and found that 18 of these genes were missing across the 35 GEMs that could not simulate growth. Essential genes were found by removing one gene at a time from the pan-STM.v1.1 reconstruction and simulating for biomass production using FBA. The removal of a gene was accompanied by the removal of the encoded metabolic reactions that were found using the find_gene_knockout_reactions tool available in the COBRA toolbox. A missing gene was classified as essential when its removal from the model resulted in a non-growth phenotype prediction. We further set out to identify potential auxotrophies caused by these gene deletions. Three GEMs that were flagged due to missing Salmonella-specific metabolic traits, were subsequently excluded from the analysis because they were found to be missing over 70 essential genes. We traced back this aberration to the nucleotide sequence and found it to contain over 200 regions of unassigned nucleic acid bases ("Ns"). We subsequently searched all genomic nucleotide sequences for a high occurrence of unassigned nucleotide bases and excluded another 27 genomic sequences from further study as described in Reconstruction of a consensus Typhimurium str. LT2 GEM.

In each flagged GEM, we iteratively knocked out one of the 18 essential genes. We then simulated the sequential addition of extracellular nutrients by setting the lower bound of the exchange for that nutrient to −50 mmol / gDW / h. If growth was achieved, the essential gene was classified as the source of a potential auxotrophy and the corresponding extracellular nutrient as the nutrient for which the strain is auxotrophic. These strain-specific auxotrophies are only hypotheses because there could be alternative pathways that are not yet known for the metabolism of the essential components. However, in two cases they pointed towards metabolic pathways that were missing in the starting reconstruction (anthranilate and NAD biosynthesis biosynthesis). When nutritional supplementation did not yield growth, and we could not find an alternative network that would compensate for the metabolic requirement, the reactions were added back as orphan reactions (Supplementary Data 6).

**Conditional gene essentiality**. The base model (pan-STM.v1.1) was used to simulate growth of all 410 GEMs on 531 nutrient conditions Supplementary Data 4) and gene essentiality sets were identified in each condition using the command cobra.flux_analysis.single_gene_deletion from the COBRApy package (Supplementary Data 7). We then subtracted from each set the genes found to be essential in glucose+M9 minimal medium. The remaining genes were classified as conditionally essential genes (CEG) and mapped to the corresponding nutrient conditions.

**Mutant fitness across hosts**. We retrieved fitness scores associated to Salmonella genes calculated by Chauduri et al. across 4 hosts including 3 food producing animals and BALB/c Mice[36,37]. Briefly, a total of ~10,000 transposon mutants were generated and combined into pools which were then introduced into the animals. Input and output pools were collected and sequenced. Each mutant's fitness score was reported as the log2-fold change between the number of sequence reads obtained across the boundaries of each transposon insertion between the input and output pools. We then filtered out the fitness scores with an adjusted p-value of <0.1 and an absolute log2-Fold change of >2.

## Data availability

All data generated in this study are included in this published article (and its supplementary information files). Models are available on BioModels with accession MODEL1807280001. All genomic sequences analyzed in this study are publicly available on PATRIC[16] and NCBI[15] and accession numbers are available in Supplementary Data 1.

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

## Acknowledgements

This research was supported by the Novo Nordisk Foundation through the Center for Biosustainability at the Technical University of Denmark (NNF10CC1016517), and the NIH NIAID grant (1-U01-AI124316-01).

## Author contributions

Y.S., B.O.P. and J.M.M. designed research; Y.S. collected data, performed research, and analyzed and interpreted the data; J.M.M. and Y.S. drafted the article; Y.S., E.K., J.L., E.C., X.F., J.M.M., B.O.P., M.R. and S.N. critically revised and approved the article.

## Additional information

**Competing interests:** The authors declare no competing interests.

