## [Peer Review File · Nature Communications]

Reviewers' Comments:

Reviewer #1:

Remarks to the Author:

In the manuscript entitled " Genome-scale metabolic reconstruction of multiple Salmonella strains reveal serovar - specific metabolic traits" the authors describe the construction of metabolic network models for a variety of different Salmonella serotypes, which allows them to distinguish those serotypes based on the metabolic properties and furthermore allows them to make predictions about the metabolic properties (auxotrophies) of those serotypes. I find the overall approach interesting and the findings nicely presented. There has been a paucity of studies explicitly comparing predictions from genome-scale models with experimental measurements, which is absolutely for improving our models. I especially like the complete and easy to follow presentation in the methods section in combination with a lot of the raw 'data' presented as tables.

I have to first point out that I am not an expert in metabolic network construction and flux balance analysis, so some of my questions may seem rather amateurish.

Major points:

1) I think that the paper would benefit from a bit more discussion of the motivation and context of the work: What questions motivated the authors to do this study? What do the findings mean? Many of the findings are presented as data without describing what we learn from it. In addition, the discussion section mainly repeats the findings and discusses very little their meaning in a broader context.

2) It would be nice if the manuscript clarify a bit more regarding how predictive the GEM- FBA based approach really is. First, in several cases (like Fig 3B or Fig5a) the findings were obtained from subsets of the original set of GEMs. It is in many cases not clear how this subset was chosen (eg why is Fig5a restricted to *S. enteritidis* strains?). This choice of subsets may influence the outcome, as the presented analysis may work for some subsets but not others. Second, what does the GEM- FBA approach really add? For example, lets say the *trpC* gene is missing. Why do I need a FBA simulation to guess that the organism is Trp auxotroph? This issue becomes even more severe since the GEMs are (as far as I understood) in the end hand-curated, thus reaction pathways are added or removed manually. This leads a little to a type of self-fulfilling prophecy, when I know from the literature that the geneX is essential for using metabolite x and put this into my model, that then tells me that upon knock out of X the bacteria cannot grow on x anymore, how surprised should I be? My guess is that there are many examples where the FBA predictions go beyond such simple "presence/absence" type predictions, and I think that these would be nice to highlight.

Third, the classifications in Fig 3b and Fig 4A show a rather good separation by metabolic properties. However, I wonder if that is something specific or every set of bacteria can be separated into two groups. Imagine you have a group of bacteria that is just randomly different in its metabolic properties concerning a high number of metabolites, one can imagine a 'separation scheme' (like in Fig 3b and Fig4a) that separates this set of bacteria into any possible pair of subsets. In other words, the fact that the bacteria can be separated into two subsets does not necessarily mean that the properties in these subsets are characteristic features of the bacteria in the subset. Is there a null model or negative control to address this?

Minor points:

* line 114: what does normalization mean? Using Prokka?

*I am not sure how much Fig 1 really adds. What is the information obtained from all these plots? The central piece of information that matters later on is that there are genomic differences between the different serovars. Could that be presented more compactly? I understand these cumulative plots allow one to add a lot of information into one plot, but is it really necessary to understand the rest of the paper? I don't want to force the authors to change their paper into directions that they are not happy with, but for me upon reading it, it was more distracting the

informative.

- * Fig1A: How should one read the phylogenetic tree? Eg are the serovars that are hit by the lines of the tree that represent *S. enterica enterica* those species or is it the whole area between these lines?
- * Fig1B The gamma value for the Salmonella case is 0.629 for all the Salmonella is 0.512 (Fig S2) which is more similar to Fig1B right panels. This shows that the sampling for Fig 1B lower left panel, had a strong influence on the gamma value and makes me wonder how comparable the different gamma values in Fig 1B are. It makes me even more wonder how much the sampling of subgroups throughout the rest of the paper influences the outcomes (see above).
- * Fig1D is not explained in the text or caption.
- * what are the black stripes in Fig 1C?
- * Fig2A: I feel it might be easier to show not the total number of reactions but the percentage of core vs accessory and write the total number behind it (just a suggestion).
- * Fig 3A: What are the nutrients?
- * Fig 4A yes and no are flipped compared to Fig3A.
- * Fig 4B is not referenced in the text and I do not understand what it tells me.
- * Fig 4C shows 62 strains but the caption says 29... How were these 62 chosen?
- * Fig 4C: the legend says there are 11 extra intestinal human pathogens but there are much more in the bar plot
- * Fig 4C: the authors state that the ability to metabolize the same metabolites within a group that has the same host shows that they are adapted to the same niche. However, the total number for each host are rather low and again a subset of nutrient conditions (36 out of 323 in 532 media conditions) were picked, which could make this correlation just appear by random.
- * Fig 5a: 'number of GEMs' out of how many?
- * in line 137 the authors state: "the number of shared gene families between two Salmonella isolates decreases as the phylogenetic distance between them increases". Where can this be seen in the data?
- * line 160 the numbers for the gamma factors are different in text and Fig 1
- * line 162 what does expansion mean and how is such an expansion 'driven'?
- * line 588: unpaired
- *line 633 what is this new biomass objective function, why was it changed?
- * are negative exchange rates uptake?
- * is no growth in general if growth is below 0.001? It would be nice to have this in the main text

Reviewer #2:

Remarks to the Author:

Manuscript by Seif et al. describes prediction of metabolic capabilities of different Salmonella serovars, with the main focus on differentiating narrow-host specialists serovars from broad-host generalists serovars. The authors find that specialists tend to lose certain metabolic pathways that are retained in generalists. They also can predict auxotrophy based on the genetic make-up and confirm these phenotypes experimentally.

While the findings are likely to contribute to our overall understanding of Salmonella ecology and, maybe, pathogenesis, they are not very surprising and still inconclusive. The authors' main claim that the metabolic differences are the key in the host adaptation of Salmonella is rather unwarranted based on the current findings that only provide associative but not causative evidence.

It had been already shown that host-adapted serovars tend to evolve by gene inactivation, by far limited to the genes involved in metabolism. Thus, the findings are not particularly surprising. While the authors claim that the loss of metabolic pathways could lead to host adaptation, they do not prove it in any way. They also do not elaborate much whether such loss is reductive in nature (loss of unused traits due to the lack of metabolites in specific hosts) or how it could be adaptive (gain

of fitness to compare with generalists). There is also no discussion on the host differences that could drive such evolution. Finally, the authors do not attempt to explain how metabolic differences could lead to the difference in the pathogenesis mechanism of specialists serovars (that tend to cause systemic invasive infections) and generalists (that tend to cause localized non-invasive infections).

It is unclear from the data whether the fact that birds-adapted Pyllorum and Gallinarum serovars as well as human-adapted Typhi and ParatyphiA serovars have distinct metabolic profiles is due to independent convergence or merely their very close genetic relatedness.

Reviewer #3:

Remarks to the Author:

Seif et al. compare the genomic content of 410 Salmonella strains, and construct models to assess the metabolic diversity between strains. The authors extend the size of the pan genome, and find that the metabolic genes that are most variable are involved in inner membrane transport, carbohydrate metabolism and cell wall metabolism. Metabolic models are used to identify metabolic capabilities that distinguish serovars and different levels of host specialization.

A great deal of interesting data is provided in this paper, however the novel findings need to be more clearly delineated. For example, in line 83 it is explained that many metabolic signatures of serovars are already known. It is currently unclear the extent to which the current study simply recapitulates previous knowledge with a bigger data set, versus expands knowledge. Further, it is concerning that one of the metabolic traits that is identified as distinguishing phenotypes is inaccurately predicted when compared to experimental data.

The manuscript would also benefit from greater attention to detail. Many of the figure legends do not adequately describe the images. More concerning, the text does not match the data for the discrepancy between predicted and observed growth. The text says that discrepancies were spread across 2 of 6 strains, but the supplemental table suggests that the discrepancies are in 3 of 6 strains. This is a small error that does not substantively change the interpretation, but generates concern none-the-less.

Minor comments:

Line 155 - Are these differences in the factor significant? As noted below the error estimates appear to at least overlap.

Line 186 - This sentence is not clear to me. What is the percentage that is reported? Is the sum of 240 and 350 bigger than the total number of accessory reactions (433) because some reactions are double counted in each category?

Line 242 - I don't understand this sentence.

Line 258 - Why were the 19 media conditions chosen to display. What are the media conditions?

Line 260 - It would be useful to know how many independent origins of specialists there have been. If there has been only one then this pattern could be driven by phylogenetic non-independence, rather than selection based on environment.

Line 296 - By lethal phenotype do you mean no-growth?

Line 309 - The data in table S18 actually shows that the false prediction of growth on serine was in 439843.8 not in CVM19633. Further it is concerning that one of the primary growth phenotypes

that was identified to distinguish serovars, growth on glycolate, is incorrectly predicted by the model.

Figure 1 - In panel B what is the estimate of error that is shown? Given this estimate is there actually any significant difference between the factor? In panel C it appears that the left column often has gene families that are described as unique to one of the three serovars. In panel D what are the black numbers outside the colored circles?

Figure 3 - Panel B is discussed before panel A in the text.

Figure 4 - Panel B is not informative. In panel C if 29 strains were compared why are more than 60 strains shown in the figure?

Reviewer #4:

Remarks to the Author:

Overall evaluation

The manuscript by Seif and coworkers reports the construction of genome-scale metabolic models for 410 Salmonella strains. While the authors need to be commended for undertaking this Herculean community effort, it resulted in very few concrete novel insights, thus limiting the potential impact of the analysis because it advances the field only incrementally (specific points 1-6). One way to increase the impact of the study would be to test one of the more novel in silico predictions experimentally (specific point 4), however, this would require extensive additional experimentation.

Specific points

1) Lines 109-171: The definition of core and pan genomes expands on previous work in this area, but does not lead to any conceptual advance.

2) Lines 173-197: Reconstruction of genome-scale metabolic models reveals that Salmonella serovars differ in O-antigen biosynthesis, alternate carbon metabolism, the glyoxylate cycle, periplasmic transport and several catabolic pathways. Unfortunately, these results appear to provide no insights into or predictions on how metabolic differences explain differences in the biology of different Salmonella serovars as none are mentioned.

3) Lines 199-219: While it is reassuring that reconstruction of genome-scale metabolic models predicts the outcome of biochemical reactions used for identification of Salmonella serovars, this result provides little new insights into the biology of these pathogens.

4) Lines 221-283: Perhaps the most interesting inference from this analysis is that Salmonella serovars differ in their metabolic capabilities, suggesting that each occupies a different ecological niche. However, these predictions remain rather elusive, as the analysis lacks the depth to reveal how any of these metabolic pathways would alter host-pathogen interaction or host range. Exploring one of these hypotheses experimentally using an animal model would greatly enhance the significance of this work, whereas the sole reliance of in silico predictions severely limits the studies impact.

5) Lines 285-319: While it is reassuring that reconstruction of genome-scale metabolic models can be validated by determining growth on different carbon sources, these control experiments do not constitute a conceptual advance per se. What is the biological significance of possessing or lacking a pathway for any of the carbon sources tested?

6) Lines 321-358: While it is encouraging that genome-scale metabolic models predict autotrophies, the significance of this finding remains unclear. How do autotrophies alter the outcome of host microbe interaction or aid in occupation of a biological niche?

Point-by-Point Response

Reviewer #1 (Remarks to the Author):

- 5 In the manuscript entitled “ Genome-scale metabolic reconstruction of multiple Salmonella strains reveal serovar - specific metabolic traits” the authors describe the construction of metabolic network models for a variety of different Salmonella serotypes, which allows them to distinguish those serotypes based on the metabolic properties and furthermore allows them to make predictions about the metabolic properties (auxotrophies) of those serotypes. I find the overall approach interesting and the findings nicely presented.
- 10 There has been a paucity of studies explicitly comparing predictions from genome-scale models with experimental measurements, which is absolutely for improving our models. I especially like the complete and easy to follow presentation in the methods section in combination with a lot of the raw 'data' presented as tables.
- 15 I have to first point out that I am not an expert in metabolic network construction and flux balance analysis, so some of my questions may seem rather amateurish.

Major points:

- 20 1) I think that the paper would benefit from a bit more discussion of the motivation and context of the work: What questions motivated the authors to do this study? What do the findings mean? Many of the findings are presented as data without describing what we learn from it. In addition, the discussion section mainly repeats the findings and discusses very little their meaning in a broader context.

25 We thank the reviewer for these comments. In our rewrite we have added additional context and biological interpretation of our findings to the discussion section (lines 480-503). In this study, we investigate metabolic capabilities that are linked to an isolate's lifestyle by using genome scale metabolic modelling in concert with publically available genomic sequences. In particular, we discuss: 1) that the pan genome size of a serovar does not correspond to its host range, 2) that host-associated serovars tend to lose certain catabolic capabilities of nutrients that are readily available in the gut, and 3) that certain catabolic pathways which are

30 differentially lost across host-specific serovars are important for fitness during intestinal infection of various hosts (including pig, cattle and chicken) but not during extraintestinal infection of mice (see fig. 6 and newly added section entitled “Catabolic capabilities differentially affect fitness across a number of hosts.”).

- 35 2) It would be nice if the manuscript clarify a bit more regarding how predictive the GEM- FBA based approach really is.

40 Strain-specific metabolic network reconstructions have proven to be powerful tools to probe the effect of genomic diversity between strains of *E.coli* and *S. aureus*^{1,2}. We agree that the GEM's predictive power needs to be made clear to the reader. In this resubmission, we included a “validation section” (lines 297-336) in which we compare the predictions made for the strain-specific catabolic profiles against experimental data-sets. We compared a total of 807 of our predictions against experimentally derived data-sets and obtained 80.7% agreement overall. We feel that this level of predictability is high enough to make our downstream analyses valid. We also highlighted the failure cases in the supplementary file section 2 and added the corresponding discussion in the supplementary data File 1. The full data is available in Tables S17 and S18.

45 First, in several cases (like Fig 3B or Fig5a) the findings were obtained from subsets of the original set of GEMs. It is in many cases not clear how this subset was chosen (eg why is Fig5a restricted to *S. enteritidis* strains?). This choice of subsets may influence the outcome, as the presented analysis may work for some subsets but not others.

50

We understand that visualizing subsets of the data may confuse readers. Our decision to do this was motivated by concerns of clarity and conciseness. In the revised manuscript, we updated the figure captions to communicate these decisions to the reader. Of important note, all of the data is available as Supplementary Information (Tables S9 and S10). Figure 3B (previously 3A) only shows the results for the 54 serovars known to infect more than one host (termed generalists). And Figure 3A (previously 3B) shows a classification tree that only includes the serovars that are represented by serovars with more than 5 genomes available. Figure 5A lists all of the auxotrophies predicted using our workflow. For example, potential auxotrophies were found across 16 strains of Enteritidis, 7 strains of Paratyphi A, etc..

55

60

Second, what does the GEM- FBA approach really add? For example, lets say the *trpC* gene is missing. Why do I need a FBA simulation to guess that the organism is Trp auxotroph? This issue becomes even more severe since the GEMs are (as far as I understood) in the end hand-curated, thus reaction pathways are added or removed manually. This leads a little to a type of self-fulfilling prophecy, when I know from the literature that the geneX is essential for using metabolite x and put this into my model, that then tells me that upon knock out of X the bacteria cannot grow on x anymore, how surprised should I be? My guess is that there are many examples where the FBA predictions go beyond such simple "presence/absence" type predictions, and I think that these would be nice to highlight.

65

70

The process of reconstructing strain-specific genome scale models is explained in methods sections 2A, 2B, 2C, 3A and 3B. In the revised manuscript, we added the following definition for genome scale metabolic models: "A GEM is a curated, structured knowledge base that contains all of the biochemical transformations occurring in a cell along with a mapping of the gene encoding them ³. (methods 2.A)" Briefly, we started from an existing GEM and built strain-specific GEMs by removing content when a gene loss was observed. We used sequence homology to determine the presence/absence of a gene. This process is semi-automated and guided by the strain's observed genetic material.

75

The strain-specific GEMs were then used to simulate for the production of biomass components using flux balance analysis (methods 3.A). Flux balance analysis is a method to analyze the flow of metabolites through a metabolic network ⁴. A GEM cannot simulate growth if one of the biomass precursors cannot be produced which occurs when there is a gap in one of the anabolic pathways. In this way, GEMs remove the guesswork from searching for gene disruptions and instead evaluate the effect of a group of missing genes on the metabolic network at the systems level. All phenotypes are calculated based on the possible biochemical reactions encoded within the genome. While simple gene knockouts may be easy to infer from genome alone, genome-scale models can evaluate the effect of multiple gene knockouts that may not be immediately obvious (e.g. in the case of synthetic lethal genes). Additionally, reactions are only added to a reconstruction when there is a clear genetic basis (which is why we looked for homology of metabolic genes across all 410 strains (see Table S5). Thus, the strain-specific models are a metabolic representation of the genomic sequences of naturally occurring strains.

80

85

Third, the classifications in Fig 3b and Fig 4A show a rather good separation by metabolic properties. However, I wonder if that is something specific or every set of bacteria can be separated into two groups. Imagine you have a group of bacteria that is just randomly different in its metabolic properties concerning a high number of metabolites, one can imagine a 'separation scheme' (like in Fig 3b and Fig4a) that separates this set of bacteria into any possible pair of subsets. In other words, the fact that the bacteria can be separated into two subsets does not necessarily mean that the properties in these subsets are characteristic features of the bacteria in the subset. Is there a null model or negative control to address this?

90

95

We are only beginning to understand the metabolic properties of certain less-well-studied serovars, and are attempting to accelerate this process with genome-scale modeling. These results may serve for the design of new assays for the identification of serovars. In Fig 4A, we make the observation that strains known to be restricted to one host tend to lose catabolic capabilities. There are several differences in catabolic capabilities but the decision tree helps by determining the most discriminating capabilities with the minimal amount of information. In the revised manuscript, we added measures of impurity (gini) to the figures (Fig 3 & Fig 4) which represents the probability of misclassification of a set of instances. We are confident in our separation scheme

100

here based on the number of strains and the lack/presence of capabilities across strains that have been observed in previous studies^{5,6}.

Minor points:

105 * line 114: what does normalization mean? Using Prokka?

110 One of the false predictions that may arise when comparing multiple genomes comes from differential gene calling across various annotation platforms. As such, “normalizing” the genome annotation means that genes were called using the same annotation platform, in this case Prokka, (and underlying algorithm) across all of the genomes. However we agree that the terminology may confuse readers. In the revised manuscript, we have replaced the text (lines 115-116) to read: “All genomes were re-annotated using a consistent annotation platform to avoid differential gene calling.”

115 *I am not sure how much Fig 1 really adds. What is the information obtained from all these plots? The central piece of information that matters later on is that there are genomic differences between the different serovars. Could that be presented more compactly? I understand these cumulative plots allow one to add a lot of information into one plot, but is it really necessary to understand the rest of the paper? I don't want to force the authors to change their paper into directions that they are not happy with, but for me upon reading it, it was more distracting than informative.

120 In Figure 1, we have introduced the concept of a serotype-specific core and pan genome. It is the first time that such a concept is presented (to our knowledge) on such a scale and is utilized to compare serotype-specific pan genomic features. This figure highlights the fact that there are pan-genomic features that are specific to each serovar. These features are interesting because they highlight the extent of genomic differences between serovars of *Salmonella*.
125 For example, we find that a small pan genome (i.e. variation in genomic content) does not necessarily correspond to a restricted range of host. This observation is important because it contradicts one of the previous beliefs that a larger pan genome is indicative of a larger host range (lines 140-147 and lines 446-454). Further, it is essential for this discussion because we later proceed to hypothesize that one of the key differentiating factors are the catabolic capabilities of the strain conferred by the genes. So in other words, it's not the number of gene families contained in
130 the pan genome that is important but rather the specific function of the gene families that a strain carries in its genome. We use GEMs and FBA to predict the effect of genome-reduction on strain specific phenotypes - this has never been done before for *Salmonella* on such a large scale.

135 * Fig1A: How should one read the phylogenetic tree? Eg are the serovars that are hit by the lines of the tree that represent *S. enterica enterica* those species or is it the whole area between these lines?

140 The whole area in between the lines represent *S. enterica* subspecies *enterica*. In order to avoid confusing the readers, we modified the figure so that the information is represented more clearly and the strains of *S. enterica* were represented in black. We also would like to note that this plot does not represent a phylogenetic tree but that we added a strain's phylogeny to help the reader contextualize the information. We added this note in the caption for figure 1A in the revised manuscript.

145 * Fig1B The gamma value for the *Salmonella* case is 0.629 for all the *Salmonella* is 0.512 (Fig S2) which is more similar to Fig1B right panels. This shows that the sampling for Fig 1B lower left panel, had a strong influence on the gamma value and makes me wonder how comparable the different gamma values in Fig 1B are. It makes me even more wonder how much the sampling of subgroups throughout the rest of the paper influences the outcomes (see above).

150 We agree that sampling as well as the number of included genomic sequences can affect the fit of Heap's Law. In figure S2, all genomes in the data set ($n = 410$) were sampled and the median gamma parameter was reported while in figure 1B the pan genome curve was cut at addition 41 and heap's law was fitted to the

155

reduced curve. In order to identify the sensitivity of the Heap's law parameters, we fitted Heap's law to pan genome curves built using 10, 20, 30 and up to 410 genomes. We found that the averaged gamma factor decreases with a growing number of strains (see figure below, supplementary File 1). In our revised manuscript, we decided to exclude gamma values in the main text as they do not seem to reliably report a measure by which we can compare pan genome curves. Instead, we opted to report the average and standard deviation of the number of gene families in the pan genome when the 20th genomic sequence is added (see methods 1.D).

160

165

* Fig1D is not explained in the text or caption.

In our revised manuscript, we have added an explanation to the caption of Figure 1; "We identified the serovar-specific core gene families for serovars Paratyphi A, Typhimurium and Enteritidis and plotted a Venn diagram to represent the shared content. Paratyphi A has the highest number of core gene families, of which 1,014 are not part of the Typhimurium nor the Enteritidis core genome."

170

* what are the black stripes in Fig 1C?

175

As highlighted in the legend, the areas filled in dark blue appear as black strokes because they represent the presence of a gene family in a certain genome. Because the clustermap shows a total of 2,526 gene families, when the cluster of gene families is not large, the presence of gene families only appears as black strokes. The full information is available in Table S15.

180

* Fig2A: I feel it might be easier to show not the total number of reactions but the percentage of core vs accessory and write the total number behind it (just a suggestion).

185

* Fig 3A: What are the nutrients?

The nutrients are shown in figure S4, a reference to that figure was added in the figure caption in our rewrite.

* Fig 4A yes and no are flipped compared to Fig3A.

190

In order to be consistent, we flipped the order in Fig. 4A.

* Fig 4B is not referenced in the text and I do not understand what it tells me.

195

This figure was meant to convey the general idea behind the analyses. However, because it is not very informative, we decided to remove it.

* Fig 4C shows 62 strains but the caption says 29... How were these 62 chosen?

200

The caption was modified to amend for this mistake. There were actually 67 strains (including all Paratyphi A strains) that were chosen based on their classification as “specialists” in the literature, meaning that they have been explicitly shown to be restricted to one host.

* Fig 4C: the legend says there are 11 extra intestinal human pathogens but there are much more in the bar plot

205

This legend was mistakenly annotated. There are 41 Paratyphi A strains in the data-set (as shown in figure 1B and available in Table S5) plus an additional 6 Typhi strains and 1 Paratyphi C strain. The reviewers are correct in pointing this error out. In the revised manuscript, we modified the legend to accurately reflect the data.

210

* Fig 4C: the authors state that the ability to metabolize the same metabolites within a group that has the same host shows that they are adapted to the same niche. However, the total number for each host are rather low and again a subset of nutrient conditions (18 out of 323 in 532 media conditions) were picked, which could make this correlation just appear by random.

215

The subset was picked to highlight all of the differences in catabolic capabilities found across the data-set. In other words, the catabolic capabilities that are shared across all of the strains were excluded. Unfortunately, we are limited in the data and knowledge in host specificity that we have. Loss/gain of ability to catabolize even a single nutrient has been shown to have wide ranging effects on pathogenicity and epidemiology ⁷, we feel that 18 such predictions are actually quite significant. A similar prediction (in which galactarate catabolism was identified as a capability that differentiates intrainestinal strains from extraintestinal strains) lead to the identification of the mechanism by which gastrointestinal *Salmonella* thrived after the administration of streptomycin ⁸.

220

* Fig 5a: 'number of GEMs' out of how many?

225

There are varying numbers of GEMs that correspond to each serovars in this study. Figure S1, represents graphically the number of genomes per serovars. Additionally, table S6 lists out metadata for each GEM. In order to make the information more easily accessible, we also added the number of GEMs per serovar listed out in the modified figure.

230

* in line 137 the authors state: “the number of shared gene families between two *Salmonella* isolates decreases as the phylogenetic distance between them increases”. Where can this be seen in the data?

235

To fortify this statement we ran an additional analysis in which we extracted the 7 housekeeping genes known for *Salmonella* and obtained phylogenetic distances by aligning them. Then, for each pair of genomes, we computed the number of shared gene families. We proceeded to calculate the pearson correlation between the

240

phylogenetic distance and the number of unshared gene families and found a correlation of 0.851 (p-value < 0.01). We have added those results as a supplementary figure (supplementary data File section 1 and Figure S3). This analysis demonstrates that the number of shared gene families between two strains decreases as phylogenetic distance increases.

245 * line 160 the numbers for the gamma factors are different in text and Fig 1

In the revised manuscript, the method for fitting Heap's law to the pan genome curves was removed. We used instead the average and standard deviation of the number of gene families at the 20th addition to quantitatively compare pan genome content across serovars.

250

* line 162 what does expansion mean and how is such an expansion 'driven'?

We understand that such terminology may be elusive. A pan genome is 'expanded' when there is a larger rate of appearance of new gene families. In order to avoid further confusion we modified the text to read: " *S. Paratyphi A* and *S. Enteritidis* have a similar number of gene families at addition 20 ($p(20) = 4,527 \pm 43$ and $p(20) = 4,606 \pm 75$ respectively). Additionally we found that the *S. Typhimurium* pan-genome is as large as the *Salmonella* pan-genome ($p(20) = 6,559 \pm 273$ and $p(20) = 7,676 \pm 711$, respectively), suggesting that strains of this serovar are a major source of *Salmonella* gene content variation" (lines 160-167)

255

260 * line 588: unpaired

Noted, the word was corrected.

*line 633 what is this new biomass objective function, why was it changed?

265

The biomass function was only modified in that the O-antigen was removed from the list of biomass precursors that need to be synthesized to achieve growth. It was changed because O-antigens vary greatly in structure and biosynthetic pathways across different serovars and these biosynthetic pathways are not completely characterized.

270

* are negative exchange rates uptake?

Yes, and thank you for pointing that out. It might confuse readers. We added a clarification was in methods section 3A.

275

* is no growth in general if growth is below 0.001? It would be nice to have this in the main text

Yes, levels of growth below 0.001 are generally considered numerical errors that arise due to imperfect linear programming solver accuracy. We have added a clarification to manuscript to address this (section 3B).

280 Reviewer #2 (Remarks to the Author):

Manuscript by Seif et al. describes prediction of metabolic capabilities of different *Salmonella* serovars, with the main focus on differentiating narrow-host specialists serovars from broad-host generalists serovars. The authors find that specialists tend to lose certain metabolic pathways that are retained in generalists. They also can predict auxotrophy based on the genetic make-up and confirm these phenotypes experimentally. While the findings are likely to contribute to our overall understanding of *Salmonella* ecology and, maybe, pathogenesis, they are not very surprising and still inconclusive.

285

We appreciate the reviewer's close reading of our manuscript and their comment regarding our results contributing to the overall understanding of *Salmonella* ecology and metabolic capabilities. We would like to point out that these results systematically elucidated the major gains/losses of catabolic capabilities across the *Salmonella* species. In this resubmission we have provided further evidence to support our claim that certain catabolic capabilities contribute differentially to fitness across different hosts and that the major losses across host-restricted serovars coincide with these catabolic pathways (see section entitled "Catabolic capabilities differentially affect fitness across a number of hosts"). While some of the results may not be surprising because they recapitulate previous findings, our manuscript presents several exciting hypotheses that implicate certain metabolic capabilities as being beneficial for the colonization of different hosts. These hypotheses are listed so they can be readily addressed in future studies that we hope will lead to even better understanding of *Salmonella* ecology.

290

295

300

The authors' main claim that the metabolic differences are the key in the host adaptation of *Salmonella* is rather unwarranted based on the current findings that only provide associative but not causative evidence.

305

We understand how the phrasing in line 434 could lead the reader to believe that our claim is that metabolic differences are the sole consequence of host adaptation. In the revised manuscript, we modified the sentence to read the following: "Our study demonstrates that strain-specific models of *Salmonella* metabolism can be used to systematically identify a serovar's unique metabolic capabilities." (lines 513-517)

310

Providing direct causative evidence for certain metabolic differences as being key to host adaptation is an active field of investigation today. With this study we first attempt to provide links between a strain's metabolic capabilities and its genotype, and then validate some of these links using experimentally observed growth phenotypes. We subsequently compare the strain-specific metabolic capabilities amongst strains whose host range is known and use it to guide the generation of hypotheses. While we do not test these hypotheses explicitly in this study, there are other studies that support our predictions and provide causative evidence of a given strain's host range linked to its ability to catabolize a given nutrient. Further, we took one additional step to identify candidate metabolites that confer fitness to *Salmonella* infection in different host environments (See "Catabolic capabilities differentially affect fitness across a number of hosts", lines 374-435 and figure 6). Follow-up studies are indeed warranted to confirm the hypotheses postulated here, however such studies can be lengthy and we do not feel that they are warranted for the scope of this paper.

315

320

It had been already shown that host-adapted serovars tend to evolve by gene inactivation, by far limited to the genes involved in metabolism. Thus, the findings are not particular surprising.

325 In addition to metabolism, gene inactivation accumulates in a range of other genes, including those involved in
adhesion, motility, effector proteins, etc. The overall goal of our study was to model the metabolism of these
organisms and highlight losses that may be linked to host range. The added value of this workflow is that it
presents a systematic accounting of the genes that have been lost in each strain linked to their phenotypes and
nutrient usage ability. To our knowledge, no others have performed such a large and systematic analysis of
330 gain/loss of phenotypes across *Salmonella* strains. Our analysis presents a set of direct, testable hypotheses
for the community about functional differences between strains of *Salmonella*.

While the authors claim that the loss of metabolic pathways could lead to host adaptation, they do not prove it
in any way.

335 In the revised manuscript, we have analyzed data that were collected from animal studies performed by others.
Specifically, we analyzed a set of gene knockout fitness studies performed in 4 different animal hosts (pigs, cattle,
chicken, and mice). Many of the catabolic genes that differentially affect fitness across different hosts correspond
to the genes we identified as missing in subsets of host-restricted serovars (lines 409-435). Based on this validation, we
have discussed clear hypotheses regarding how specific metabolic capabilities may affect host range and the
340 preferred nutritional niche of different serovars. Determining factors which affect host range is a problem that has
been under active investigation for decades. Finding the specific phenotype or environment where a metabolic
capability impacts host range has been very challenging for the field (e.g., *Salmonella* has a broad host range and a
phenotype may only manifest itself in a specific host species under specific conditions). Moreover, it can take several
years to experimentally realize such predictions due to the necessity of performing detailed animal work.
345 Nonetheless, our study highlights interesting trends that may help shed light on this difficult area. We provide a
roadmap for experimental work and specific predictions that are meant to stimulate future experiments.

They also do not elaborate much whether such loss is reductive in nature (loss of unused traits due to the lack
of metabolites in specific hosts) or how it could be adaptive (gain of fitness to compare with generalists). There
350 is also no discussion on the host differences that could drive such evolution.

We thank the reviewers for pointing out ways to improve our manuscript. To address these concerns, we highlighted
a few examples of host differences that could have driven the loss of certain genes and, by extension, catabolic
pathways across host-adapted serovars (e.g. Typhi, Paratyphi A and Choleraesuis). Please refer to the modified
355 results sections highlighted in red for a more detailed answer to this concern (lines 491-503 and supplementary File 1
section 5).

Finally, the authors do not attempt to explain how metabolic differences could lead to the difference in the
pathogenesis mechanism of specialists serovars (that tend to cause systemic invasive infections) and generalists
360 (that tend to cause localized non-invasive infections).

In the revised manuscript, we added a section entitled “Catabolic capabilities differentially affect fitness across a
number of hosts.” This section describes how we found that catabolic genes that were lost amongst host restricted
serovars contribute to fitness during intestinal infection of various hosts but not splenic infection of mice. In the
365 section entitled “GEMs enable investigation into the genetic basis of serovar-specific auxotrophies”, we highlighted a
number of GEM-predicted auxotrophies across several *Salmonella* strains (lines 344-372). Divergent nutrient
utilization capabilities as well as nutrient auxotrophies have revealed events of pathoadaptation and have been
shown to enhance virulence in strains of *Shigella*⁹, *C. difficile*⁷, and *Salmonella*⁸. In the revised manuscript we
expanded the discussion section with one example describing a metabolic capability that confers an advantage to
370 *Salmonella* strains (lines 481-489) as well as an additional figure reprinted below:

Figure 6: Conditionally essential genes (CEGs) and corresponding mutant fitness in diverse hosts. **A)** We searched for CEGs across 531 nutrient environment conditions and found a total of 20 involved in central metabolism. Of the 531 nutrient environments, 242 were anaerobic and 289 were aerobic. We plot here the 10 most frequent CEGs in aerobic and anaerobic conditions. **B)** We selected for CEGs whose corresponding mutant was found to contribute to fitness in at least one host. We then plotted a venn diagram of CEGs that were observed to be important for fitness in intrainestinal versus extraintestinal hosts. We subdivided CEGs into those that were predicted to be essential on 5 or more nutrient environments (mCEGs) and those predicted to be essential in less than 5 nutrient environments (sCEGs). **C)** We identified the occurrence of sCEGs that were seen to contribute to fitness across strains of *Salmonella*. We highlighted here three catabolic pathways featuring the selected sCEGs and the number of *Salmonella* strains that do not carry the CEGs in their genome. The genes are placed next to the metabolic process that they are involved in. The fitness contributions of an sCEG to hosts are highlighted (ovals indicate that a significantly affected fitness was measured in this host).

375

380

385 It is unclear from the data whether the fact that birds-adapted *Pyllorum* and *Gallinarum* serovars as well as human-adapted *Typhi* and *Paratyphi A* serovars have distinct metabolic profiles is due to independent convergence or merely their very close genetic relatedness.

390 As a measure of phylogenetic distance, we constructed a phylogenetic tree by aligning the 7 concatenated *Salmonella* housekeeping genes (including *aroC*, *dnaN*, *hemD*, *hisD*, *purE*, *sucA* and *thrA*¹⁰) across all strains. We found that while strains of several serovars (for example *Cerro*, *Thompson* and *Agona*) are phylogenetically more closely related to strains of *Gallinarum* than *Pullorum* strains are, they do not share similar catabolic capability profiles. Additionally, while strains of *Typhi* and *Paratyphi A* are closely related, we observed differential loss of conditionally essential genes (CEGs). For example, utilization of galactonate as a sole carbon source is predicted to be lost amongst both *Typhi* and *Paratyphi A* strains. But while *Paratyphi A* strains experience loss of *dgoK*, *Typhi* strains lost *dgoA*, *dgoK* and *dgoT*. Zhou et. al argues that gene loss has occurred early in the history of *Paratyphi A*¹¹. We hypothesize that loss of *dgoA* and *dgoT* may have occurred after serovar diversification and could have been driven by non-essentiality of galactonate transport in the human extra-intestinal environment and represents a form of convergent evolution. We added these analyses to the supplementary data File 1 (section 4).

400

Reviewer #3 (Remarks to the Author):

405 Seif et al. compare the genomic content of 410 *Salmonella* strains, and construct models to assess the metabolic diversity between strains. The authors extend the size of the pan genome, and find that the metabolic genes that are most variable are involved in inner membrane transport, carbohydrate metabolism and cell wall metabolism. Metabolic models are used to identify metabolic capabilities that distinguish serovars and different levels of host specialization.

410 A great deal of interesting data is provided in this paper, however the novel findings need to be more clearly delineated. For example, in line 83 it is explained that many metabolic signatures of serovars are already known. It is currently unclear the extent to which the current study simply recapitulates previous knowledge with a bigger data set, versus expands knowledge.

415 In the revised manuscript, we have listed the known metabolic signatures for serovars in lines 466-471 and all of the novel findings in the section entitled “GEM predicted growth capabilities differentiate host-restricted strains from strains capable of colonizing a broad host range” as well as in figures 3. We used strain-specific metabolic signatures identified in other studies as a validation set. However, with the exception of serovar Typhi, the metabolic signatures were never tested on more than one strain of the same serovar and therefore did not account for strain to strain variation (see section entitled “Experimental validation of metabolic networks and nutrient utilization predictions shows high model accuracy”). With our findings (Fig 3A), we identify the metabolic capabilities that are predicted to be shared by more than 5 strains of the same serovar. We also delineate explicitly the novel findings in lines 270-296.

420 Further, it is concerning that one of the metabolic traits that is identified as distinguishing phenotypes is inaccurately predicted when compared to experimental data.

425 Thus far, there have been no molecular studies detailing the glycolate utilization biochemical pathway in *Salmonella*. We see that the annotated pathway in the *Salmonella* GEM mirrors the pathway that has been characterized in *E. coli*, which means that it was probably added as a result of observed sequence homology of the genes involved in the pathway. Glycolate uptake and utilization is mediated by a multi-protein complex in *E. coli* (including *glcDEF*). However, only one gene (*glcD*) was found to correspond to such a function in *Salmonella*. Additionally, while it is annotated in the model as a glycolate oxidase, its genome annotation was that of a putative lactate oxidase. We noticed that orthologs for *glcD* are only found in 46% of the genomes included in this data-set. We also found conflicting reports of the capability of several *Salmonella* strains (including str. LT2) to successfully utilize glycolate^{125,13}. Intriguingly, it was shown that STM1620 highly contributes to fitness of *Salmonella* strains across different hosts¹⁴. It is very likely that the exact function of STM1620 was not correctly annotated in pan-STM.1.v2. We have added a section in supplementary data File 1 (section 2) in which we discuss this problematic prediction. We also modified Figure 3B (now 3A) to exclude glycolate utilization as a differentiating metabolic capability because it could mislead the reader. The validation of predictions with experimental data serves the function of highlighting such knowledge gaps.

440 The manuscript would also benefit from greater attention to detail. **Many of the figure legends do not adequately describe the images.**

445 We apologize for the poor description of the figures. Closer attention to detail has been given in the revised manuscript.

450 More concerning, the text does not match the data for the discrepancy between predicted and observed growth. The text says that discrepancies were spread across 2 of 6 strains, but the supplemental table suggests that the discrepancies are in 3 of 6 strains. This is a small error that does not substantively change the interpretation, but generates concern none-the-less.

In our revised manuscript, we modified the text accordingly: “However, there were 5 failure cases across 3 strains in total (discussed in Supplementary data 1).” Further discussion of the failure cases were added in the supplementary data.

455

Minor comments:

Line 155 - Are these differences in the factor significant? As noted below the error estimates appear to at least overlap.

460

Please refer to the response to reviewer #1 (for the question regarding line 162 and fig 1B) for concerns related to the overlap of error estimates. Briefly, we agree with the reviewer and have removed the method of fitting Heap's law to assess the pan genome size.

465

Line 186 - This sentence is not clear to me. What is the percentage that is reported? Is the sum of 240 and 350 bigger than the total number of accessory reactions (433) because some reactions are double counted in each category?

470

The percentage is reported out of a total number of reactions annotated in the subsystem. We replaced this number with the number of reactions/metabolic processes in this subsystem that are part of the accessory reactome: "Metabolic processes involved in carbohydrate metabolism and inner membrane transport compose a large percentage of the accessory reactome, 62 (14.3%) and 72 (16.6%) metabolic reactions and processes, respectively (Table S11)."(lines 190-195)

Line 242 - I don't understand this sentence.

475

In the revised manuscript, we have removed the sentence.

Line 258 - Why were the 19 media conditions chosen to display. What are the media conditions?

480

The media conditions are shown in figure S4. A limited number of nutrient conditions were displayed for purposes of clarity. The nutrient conditions were selected to highlight catabolic capabilities with the largest variance across the 53 serovars. Specifically, we considered the matrix containing the predicted growth phenotypes across all strains (available in Tables S10 and S11). Then for each nutrient conditions we obtained the number of strains that were predicted to be incapable of growth and selected the top 19 nutrient conditions.

485

Line 260 - It would be useful to know how many independent origins of specialists there have been. If there has been only one then this pattern could be driven by phylogenetic non-independence, rather than selection based on environment.

490

There are a total of 4 different hosts that the specialists have been shown to be restricted to; extra-intestinal human, swine and bovine, avian, and cold blooded animal. In figure 4B, we show the niche that the specialists included in the data-set have been documented to be restricted to. More detailed information is available in Table S6. We also specified the hosts that the specialists are known to colonize throughout the text.

495

Line 296 - By lethal phenotype do you mean no-growth?

Yes, we replaced the term 'lethal' with 'no-growth' to avoid further confusion

500

Line 309 - The data in table S18 actually shows that the false prediction of growth on serine was in 439843.8 not in CVM19633.

That is because CVM19633 is the strain's name but 439843.8 is its genome accession number. We find that it is more reliable to refer to the genome accession number in this case because the predictions made are based on the genome content. In that way, the efforts made in this study are replicable. You can find a mapping between the strain name and the genome accession number in table S6.

505

Further it is concerning that one of the primary growth phenotypes that was identified to distinguish serovars, growth on glycolate, is incorrectly predicted by the model.

510 See the response to your previous comment (“Further, it is concerning that one of the metabolic traits that is identified as distinguishing phenotypes is inaccurately predicted when compared to experimental data”)

515 Figure 1 - In panel B what is the estimate of error that is shown? Given this estimate is there actually any significant difference between the factor? In panel C it appears that the left column often has gene families that are described as unique to one of the three serovars. In panel D what are the black numbers outside the colored circles?

520 Panel B: The estimate of the error shown is the standard deviation for the gamma factor obtained from fitting Heap’s law. Due to the large deviation of the fitted values across the sampled curves we decided to report the average and standard deviation of the number of gene families observed at the 20th genomic addition. We chose the 20th genomic addition in order to have a standard deviation of the average number of gene families in the Paratyphi A genome that is reflective of the true standard deviation for this serovar (for which we only have 41 genomic sequences). We observed that there is still a significant difference between the pan genome curves drawn for Typhimurium and *Salmonella* versus those drawn for Enteritidis and Paratyphi A but that the differences between the Typhimurium pan genome and the *Salmonella* pan genome are no longer significant.

525 Panel C: In fact the biggest cluster of gene families that are unique to Enteritidis (left column) are the gene families boxed in red. We notice that the cluster of gene families right above the boxed ones are also shared among Paratyphi A strains.

530 Panel D: The numbers were placed inside the venn diagram to avoid confusion. They represent the number of gene families shared between two serovars but not with the third serovar.

Figure 3 - Panel B is discussed before panel A in the text.

This panel was removed.

535 Figure 4 - Panel B is not informative. In panel C if 29 strains were compared why are more than 60 strains shown in the figure?

Panel B: Was removed

540 Panel C: That was an error, there are 41 strains of Paratyphi A (as discussed in Figure 1). The bar graph was generated from the data while the legend was modified (erroneously) at a later step.

Reviewer #4 (Remarks to the Author):

Overall evaluation

545

The manuscript by Seif and coworkers reports the construction of genome-scale metabolic models for 410 *Salmonella* strains. While the authors need to be commended for undertaking this Herculean community effort, it resulted in very few concrete novel insights, thus limiting the potential impact of the analysis because it advances the field only incrementally (specific points 1-6). One way to increase the impact of the study would be to test one of the more novel in silico predictions experimentally (specific point 4), however, this would require extensive additional experimentation.

550

We greatly appreciate the reviewer’s recognition of our efforts. We have expanded the results and discussion to include the main novel findings (such as the catabolic capabilities differentiating human-specialists and cold-adapted

555 serovars; lines 270-296). We had initially validated a large amount of the predicted growth phenotypes across strains with good general agreement (83.1%). However, to provide additional support for the hypotheses that the above highlighted catabolic capabilities provide a differential advantage across a variety of microenvironments, we have added a new analysis in which we compare our findings against the fitness phenotypes of mutants during infection of 4 hosts (lines 374-435).

560 Specific points

1) Lines 109-171: The definition of core and pan genomes expands on previous work in this area, but does not lead to any conceptual advance.

565 We have introduced the concept of a serotype-specific core and pan genome. It is the first time that such a concept is presented (to our knowledge) on such a scale and is utilized to compare serotype-specific pan genomic features. We compared them for the first time against each other and observed varying pan genome features across the different serotypes. A previously posed hypothesis suggested that the ability to migrate and colonize different hosts is correlated with a prokaryote's pan genome size¹⁵. With our dataset, we show that this is not the case (lines 149-178). Another conceptual advance comes with the subsequent analyses in which we explore the metabolic content of the accessory genome and its consequence on a strain's growth capabilities using genome scale metabolic reconstructions.

570 2) Lines 173-197: Reconstruction of genome-scale metabolic models reveals that Salmonella serovars differ in O-antigen biosynthesis, alternate carbon metabolism, the glyoxylate cycle, periplasmic transport and several catabolic pathways. Unfortunately, these results appear to provide no insights into or predictions on how metabolic differences explain differences in the biology of different Salmonella serovars as none are mentioned.

580 In the revised manuscript, we have analyzed data that were collected from animal studies performed by others. Specifically, we analyzed a set of gene knockout fitness studies performed in 4 different animal hosts (pigs, cattle, chicken, and mice). Many of the genes in these studies correspond to the genes we identified as missing in subsets of specific serovars. We find that specific subsets of genes related to catabolism of a specific nutrient are responsible for differential fitness effects across the different hosts. Based on this validation, we have discussed clear hypotheses regarding how specific metabolic capabilities may affect host range and the preferred nutritional niche of different serovars.

585 3) Lines 199-219: While it is reassuring that reconstruction of genome-scale metabolic models predicts the outcome of biochemical reactions used for identification of Salmonella serovars, this result provides little new insights into the biology of these pathogens.

590 The intent of this section is to demonstrate the validity of the semi-automated process of genome-scale metabolic model reconstruction. With this validation step we are confirming that the strain-specific models in and of themselves constitute a new tool that researchers can use to address their questions.

595 4) Lines 221-283: Perhaps the most interesting inference from this analysis is that Salmonella serovars differ in their metabolic capabilities, suggesting that each occupies a different ecological niche. However, these predictions remain rather elusive, as the analysis lacks the depth to reveal how any of these metabolic pathways would alter host-pathogen interaction or host range. Exploring one of these hypotheses experimentally using an animal model would greatly enhance the significance of this work, whereas the sole reliance of in silico predictions severely limits the studies impact.

600 Again, we thank the reviewer for turning our attention towards this point. In our revised manuscript, we ran more analyses and have added a section and the corresponding discussion to the paper. Please refer to the manuscript as well as the response added to the comment for reviewer # 2. "Finally, the authors do not attempt to explain how metabolic differences could lead [..]" above, as well as Figure 6 (added as a result of those analyses).

605

610 5) Lines 285-319: While it is reassuring that reconstruction of genome-scale metabolic models can be validated by determining growth on different carbon sources, these control experiments do not constitute a conceptual advance per se. What is the biological significance of possessing or lacking a pathway for any of the carbon sources tested?

615 The loss of a catabolic pathway for a metabolite could indicate that the metabolite is not readily available in the strain's microenvironment and that the strain has subsequently evolved to lose the capability to utilize it as a nutrient source. In the revised manuscript, we have analyzed data that were collected from animal studies performed by others. Please see our answer provided to the question posed by reviewer #2 above : "While the authors claim that the loss of metabolic pathways could lead to host adaptation, they do not prove it in any way."

620 6) Lines 321-358: While it is encouraging that genome-scale metabolic models predict autotrophies, the significance of this finding remains unclear. How do autotrophies alter the outcome of host microbe interaction or aid in occupation of a biological niche?

625 Auxotrophies may indicate cases of directed evolution to a host, and have been shown to indicate cases of patho-adaptation to a host. For example, *Shigella flexneri* strains have been found to be auxotrophic for niacin due to non-functional *nadA* and *nadB*, and the introduction of both genes results in attenuated virulence (lines 509-512)⁹. A strain is hypothesized to become dependent on a certain nutrient for successful growth when that nutrient is abundant in its microenvironment. This nutrient is either provided by the host or by other bacteria. Such cases have been reported for symbiotic organisms and pathogens.

630

1. Monk, J. M. *et al.* Genome-scale metabolic reconstructions of multiple *Escherichia coli* strains highlight strain-specific adaptations to nutritional environments. *Proc. Natl. Acad. Sci. U. S. A.* **110**, 20338–20343 (2013).
2. Bosi, E. *et al.* Comparative genome-scale modelling of *Staphylococcus aureus* strains identifies strain-specific metabolic capabilities linked to pathogenicity. *Proc. Natl. Acad. Sci. U. S. A.* **113**, E3801–9 (2016).
3. Thiele, I. & Palsson, B. Ø. A protocol for generating a high-quality genome-scale metabolic reconstruction. *Nat. Protoc.* **5**, 93–121 (2010).
4. Orth, J. D., Thiele, I. & Palsson, B. Ø. What is flux balance analysis? *Nat. Biotechnol.* **28**, 245–248 (2010).
5. Langridge, G. C. *et al.* Patterns of genome evolution that have accompanied host adaptation in *Salmonella*. *Proc. Natl. Acad. Sci. U. S. A.* **112**, 863–868 (2015).
6. Nuccio, S.-P. & Bäumler, A. J. Comparative analysis of *Salmonella* genomes

- identifies a metabolic network for escalating growth in the inflamed gut. *MBio* **5**,
645 e00929–14 (2014).
7. Collins, J. *et al.* Dietary trehalose enhances virulence of epidemic *Clostridium difficile*. *Nature* **553**, 291–294 (2018).
 8. Faber, F. *et al.* Host-mediated sugar oxidation promotes post-antibiotic pathogen expansion. *Nature* **534**, 697–699 (2016).
 - 650 9. Prunier, A.-L. *et al.* *nadA* and *nadB* of *Shigella flexneri* 5a are antivirulence loci responsible for the synthesis of quinolinate, a small molecule inhibitor of *Shigella* pathogenicity. *Microbiology* **153**, 2363–2372 (2007).
 10. Alikhan, N.-F., Zhou, Z., Sergeant, M. J. & Achtman, M. A genomic overview of the population structure of *Salmonella*. *PLoS Genet.* **14**, e1007261 (2018).
 - 655 11. Zhou, Z. *et al.* Transient Darwinian selection in *Salmonella enterica* serovar Paratyphi A during 450 years of global spread of enteric fever. *Proc. Natl. Acad. Sci. U. S. A.* **111**, 12199–12204 (2014).
 12. Fricke, W. F. *et al.* Comparative genomics of 28 *Salmonella enterica* isolates: evidence for CRISPR-mediated adaptive sublineage evolution. *J. Bacteriol.* **193**,
660 3556–3568 (2011).
 13. Quan, J. A. *et al.* Regulation of carbon utilization by sulfur availability in *Escherichia coli* and *Salmonella typhimurium*. *Microbiology* **148**, 123–131 (2002).
 14. Chaudhuri, R. R. *et al.* Comprehensive assignment of roles for *Salmonella typhimurium* genes in intestinal colonization of food-producing animals. *PLoS Genet.* **9**, e1003456 (2013).
 - 665 15. McInerney, J. O., McNally, A. & O'Connell, M. J. Why prokaryotes have

pangenomes. *Nat Microbiol* **2**, 17040 (2017).

Reviewers' Comments:

Reviewer #1:

Remarks to the Author:

The authors have addressed all of my questions and concerns. As far as I can tell this manuscript is appropriate for publication, but the other referees are closer to this field so I would be inclined to defer to their judgment.

Reviewer #2:

Remarks to the Author:

The authors had undertaken quite a bit effort to revise and improve the manuscript.

Reviewer #3:

Remarks to the Author:

I appreciate the authors attention to my concerns. The additional dads analysis is helpful. The authors may wish to change "leveragde" to "leveraged" in line 214.

Response to Referees:

*Reviewer #1

The authors have addressed all of my questions and concerns. As far as I can tell this manuscript is appropriate for publication, but the other referees are closer to this field so I would be inclined to defer to their judgment.

*Reviewer #2

The authors had undertaken quite a bit effort to revise and improve the manuscript

*Reviewer #3

I appreciate the authors attention to my concerns. The additional dads analysis is helpful. The authors may wish to change "leveragde" to "leveraged" in line 214.

Thanks for catching this error. We have changed "leveragde" to "leveraged".

We thank all of the reviewers for their detailed review of the manuscript throughout the review process and their acknowledgement of the additional efforts we made to revise and improve the manuscript. We have made the corrections requested in this iteration.